# Graph Convolutions Enrich the Self-Attention in Transformers!

**Jeongwhan Choi***
Yonsei University
jeongwhan.choi@yonsei.ac.kr

**Hyowon Wi***
KAIST
hyowon.wi@kaist.ac.kr

**Jayoung Kim**
KAIST
jayoung.kim@kaist.ac.kr

**Yehjin Shin**
KAIST
yehjin.shin@kaist.ac.kr

**Kookjin Lee**
Arizona State University
kookjin.lee@asu.edu

**Nathaniel Trask**
University of Pennsylvania
ntrask@seas.upenn.edu

**Noseong Park**[†]
KAIST
noseong@kaist.ac.kr

## Abstract

Transformers, renowned for their self-attention mechanism, have achieved state-of-the-art performance across various tasks in natural language processing, computer vision, time-series modeling, etc. However, one of the challenges with deep Transformer models is the oversmoothing problem, where representations across layers converge to indistinguishable values, leading to significant performance degradation. We interpret the original self-attention as a simple graph filter and redesign it from a graph signal processing (GSP) perspective. We propose a graph-filter-based self-attention (GFSA)[1] to learn a general yet effective one, whose complexity, however, is slightly larger than that of the original self-attention mechanism. We demonstrate that GFSA improves the performance of Transformers in various fields, including computer vision, natural language processing, graph-level tasks, speech recognition, and code classification.

## 1 Introduction

Transformers are arguably one of the best feats in the field of deep learning. They are now showing state-of-the-art performance in various fields, ranging from computer vision to natural language processing, prediction tasks on graphs, speech recognition, and so forth [77, 16, 60, 61, 19, 74, 101, 45, 27, 41, 63, 51, 59, 38, 95, 72]. Recently, there have been several studies conducted on better understanding them [25, 3, 80]; there exists a common agreement among researchers that the self-attention is one of the keys leading to the success.

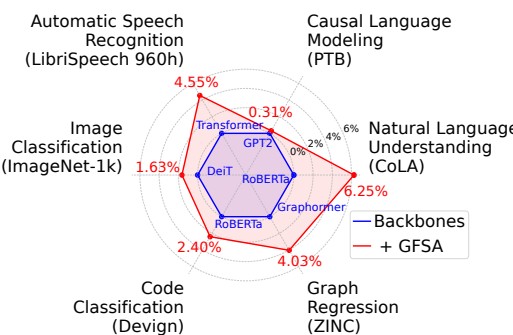

Figure 1: Performance improvements (%) of our GFSA when integrated with different Transformer backbones in various domains. We achieve these results with only tens to hundreds of additional parameters to Transformers.

---

*Equal contribution.
†Corresponding author.

[1]The source code of GFSA is available at: https://github.com/jeongwhanchoi/GFSA.

38th Conference on Neural Information Processing Systems (NeurIPS 2024).

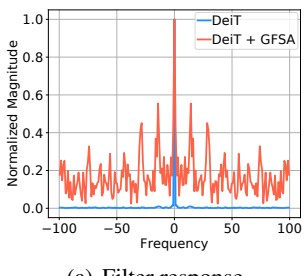 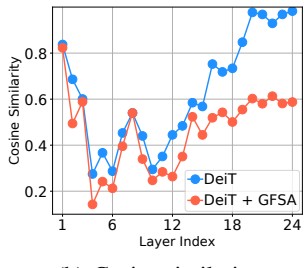 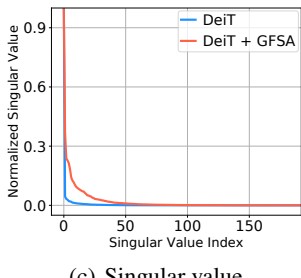

(a) Filter response

(b) Cosine similarity

(c) Singular value

Figure 2: Filter frequency response, cosine similarity, and singular values on ImageNet-1k for DeiT-S and DeiT-S + GFSA. Details and more visualizations are in Appendices C and D.

However, there also exist several studies pointing out potential limitations of the self-attention [101, 18, 25, 28]. For instance, Shi et al. [71] revealed an analogy between the self-attention and the residual graph convolutional network (GCN), showing that BERT also suffers from a notorious problem of GCNs, called *oversmoothing*, i.e., tokens' latent representations become similar to each other at the deeper layers. In every self-attention layer, value vectors are aggregated in a weighted average manner since each row-wise sum of the attention matrix is always 1. Although each self-attention layer has its own attention matrix, this aggregation method causes the oversmoothing problem, not only in Transformers but also in graph neural networks [54, 7, 79, 66, 37, 101, 25, 92, 53, 71, 80, 3, 89, 90]. However, we confine our discussion to the oversmoothing of Transformers (see Section 2).

Being inspired by them, we redesign the self-attention from the perspective of graph signal processing (GSP) — in particular, we resort to GSP on directed graphs since the attention matrix is asymmetric. However, performing graph convolutions in the self-attention layer may incur non-trivial computational overheads. Therefore, our key design point is to learn a general but effective graph filter with minimal overhead. In general, a graph filter on a graph $\mathcal{G}$ is represented by a polynomial expression based on its adjacency or Laplacian matrix — in this regard, the existing self-attention mechanism can be understood as the simplest graph filter with $\bar{A}$ only, where $\bar{A} \in [0,1]^{n \times n}$ means a learned attention matrix and $n$ is the number of input tokens.

Our proposed graph filter consists of an identity term and two matrix polynomial terms, $\bar{A}$ and $\bar{A}^K$. One can design better filters with more polynomial terms, but we avoid it since Transformers already require very large computation. The $K$-th power, $\bar{A}^K$, may also require a high computation when the number of tokens is large. To avoid this, we further approximate $\bar{A}^K$ using the element-wise first-order Taylor approximation. Therefore, one can consider that our proposed graph filter is the very next complicated filter after the one used by the original self-attention mechanism. However, its efficacy is tremendous in various fields (cf. Fig. 1).

Our proposed filter enriches the self-attention with more diverse frequency information (see Fig. 2(a)) — low (resp. high) frequency signals on $\mathcal{G}$ mean neighboring nodes have similar (resp. different) values. Therefore, our method is able to not only effectively address the oversmoothing problem but also learn better latent representations for downstream tasks.

There exist a couple of prior works to enrich the self-attention mechanism with high frequency information [80, 4]. In comparison with them, our proposed graph filter is distinctive in the following aspects: i) our proposed filter is more effective and shows better performance with comparable computational overheads, ii) our proposed filter is well-aligned with recent advancements in the GCN community — in other words, some graph filters used by recent advanced GCN methods are special cases of our proposed graph filter, which is not the case for prior works, and iii) other methods were typically studied for certain domains only whereas we test our method in 6 domains — for instance, DiversePatch [25] works only for Vision Transformers (ViTs).

We replace the self-attention layer of selected Transformers in various fields with our proposed graph filter-based layer without changing other parts. Therefore, the accuracy increases in them are solely by our proposed graph filter-based self-attention. These enriched Transformers increase the model performance by 1.63% for image classification, 6.25% for natural language understanding, 0.31% for

causal language modeling, 4.03% for graph regression, 4.76% for speech recognition, and 2.40% for code classification (see Fig. 1). Our core contributions are as follows:

- We provide a novel perspective on self-attention as a graph filter. This perspective allows us to design more effective self-attention that can address the oversmoothing problem.

- We propose a graph filter-based self-attention (GFSA) mechanism, integrating an identity term and two polynomial terms for general yet effective than the simple self-attention mechanism (Section 3).

- We demonstrate that GFSA improves the performance of Transformers on a variety of tasks. GFSA achieves improved results on natural language processing, computer vision, speech recognition, graph-level tasks, and code classification (Sections 5.1 to 5.6).

- We devise a strategy to selectively apply GFSA to even-numbered layers, effectively mitigating the computational overhead while preserving GFSA's performance (Section 6).

## 2 Background & Related Work

### 2.1 Self-Attention in Transformers

The core building block of the Transformer architecture is the self-attention mechanism, which enables the model to learn attention patterns over its input tokens [77]. The self-attention mechanism, denoted as SA : $\mathbb{R}^{n \times d} \to \mathbb{R}^{n \times d}$, can be expressed as follows:

$$\text{SA}(\boldsymbol{X}) = \text{softmax}\left(\frac{\boldsymbol{X}\boldsymbol{W}_{\text{qry}}(\boldsymbol{X}\boldsymbol{W}_{\text{key}})^{\mathsf{T}}}{\sqrt{d}}\right)\boldsymbol{X}\boldsymbol{W}_{\text{val}} = \bar{\boldsymbol{A}}\boldsymbol{X}\boldsymbol{W}_{\text{val}}, \tag{1}$$

where $\boldsymbol{X} \in \mathbb{R}^{n \times d}$ is the input feature matrix, $\boldsymbol{W}_{\text{key}} \in \mathbb{R}^{d \times d}$, $\boldsymbol{W}_{\text{qry}} \in \mathbb{R}^{d \times d}$, and $\boldsymbol{W}_{\text{val}} \in \mathbb{R}^{d \times d}$ are the key, query, and value trainable parameters, respectively, and $d$ is the dimension of each token. The self-attention mechanism allows the model to weigh the importance of each token in the input sequence relative to the others, enabling the model to capture long-range contextual information better. The Transformer architecture includes multiple layers, each with a multi-head self-attention layer followed by a position-wise feed-forward layer.

### 2.2 Self-Attention and Graph Convolutional Filter

The self-attention matrix used in Transformers has the form of symmetrically normalized adjacency matrix where each token become a node [71, 28] — the symmetrically normalized adjacency matrix is a special case of asymmetric (or directed) adjacency matrix where each row is normalized and is frequently used for the graph signal processing (GSP) on directed graphs [49]. A weighted graph $\mathcal{G}$ with adjacency matrix $\boldsymbol{A}$ can be constructed by using the input tokens as $n$ nodes and the edge weights between node $i$ and node $j$ as $\exp((\boldsymbol{X}\boldsymbol{W}_{\text{qry}})_i^{\mathsf{T}}(\boldsymbol{X}\boldsymbol{W}_{\text{key}})_j)$. We can rewrite the self-attention matrix $\bar{\boldsymbol{A}}_{ij}$ as $\frac{\exp\left((\boldsymbol{X}\boldsymbol{W}_{\text{qry}})_i^{\mathsf{T}}(\boldsymbol{X}\boldsymbol{W}_{\text{key}})_j\right)}{\sum_{k=1}^{d}\exp\left(\boldsymbol{X}\boldsymbol{W}_{\text{qry}})_i^{\mathsf{T}}(\boldsymbol{X}\boldsymbol{W}_{\text{key}})_k\right)}$. This allows $\bar{\boldsymbol{A}}$ to be interpreted as the symmetrically normalized adjacency matrix. In other words, $\bar{\boldsymbol{A}} = \boldsymbol{D}^{-1}\boldsymbol{A}$, where $\boldsymbol{D} = \text{diag}(d_1, d_2, \ldots, d_n)$ and $d_i = \sum_j \boldsymbol{A}_{i,j}$.

Our new attention method is designed on top of GSP which has a close connection to discrete signal processing (DSP) [67, 68]. In DSP, a discrete signal with a length of $n$ can be represented by a vector $\boldsymbol{x} \in \mathbb{R}^n$. Let $\boldsymbol{g} \in \mathbb{R}^n$ be a filter that we want to apply to $\boldsymbol{x}$. The convolution $\boldsymbol{x} * \boldsymbol{g}$ can be written as follows:

$$\boldsymbol{y}_i = \sum_{j=1}^{n} \boldsymbol{x}_j \boldsymbol{g}_{i-j}, \tag{2}$$

where the index, denoted as $i$, refers to the $i$-th element in each vector.

GSP can be understood as a generalized concept of DSP. Signals are defined on the nodes of a graph, and the graph's structure influences signal processing operations. In addition, the linear and shift-invariant graph convolution filter $\boldsymbol{H}$ with $n$ nodes can be written with a shift operator $\boldsymbol{S}$ as

follows — $\boldsymbol{S}$ can be from a directed graph [48]:

$$\boldsymbol{y} = \boldsymbol{H}\boldsymbol{x} = \sum_{k=0}^{K} w_k \boldsymbol{S}^k \boldsymbol{x}, \tag{3}$$

where $\boldsymbol{x} \in \mathbb{R}^n$ is a 1-dimensional graph signal, $K$ is the maximum order of polynomial, and $w_k \in [-\infty, \infty]$ is a coefficient. $\boldsymbol{S}$ is an $n \times n$ matrix where $(i,j)$-th element is non-zero if and only if there is an edge from node $i$ to $j$. Two representative samples of $\boldsymbol{S}$ are adjacency and Laplacian matrices. The graph filter $\boldsymbol{H}$ is the same as $\sum_{k=0}^{K} w_k \boldsymbol{S}^k$ with a large enough value of $K$, which is called *matrix polynomial* [48]. We note that this graph filtering operation can be extended to $d$-dimensional cases as in Eq. (1). Being inspired by Zou et al. [106] and Maskey et al. [49], we rely on the singular value domain analysis to understand the low/high-pass characteristics of filters on directed graphs (cf. Fig. 2). See more discussion in Appendices E and F.

In the context of the self-attention within Transformers, the core part of the self-attention in Eq. (1), i.e., $\bar{\boldsymbol{A}}\boldsymbol{X}$, can be considered as a $d$-dimensional graph filter with $\bar{\boldsymbol{A}}$ only, where $\boldsymbol{H} = \bar{\boldsymbol{A}}$. Our goal in this paper is to design a simple (for computational purposes) yet effective form of $\boldsymbol{H}$.

## 2.3 Oversmoothing in GCNs and Transformers

Oversmoothing is a phenomenon observed in deep learning models, especially in GCNs [39, 78]. As information is aggregated over multiple layers for multiple nodes (tokens), latent representations tend to become similar to each other, leading to a loss of distinctiveness in the representations [54, 104, 66].

Surprisingly, an oversmoothing-like phenomenon is also observed in Transformers [80, 71]. Unlike CNNs, Transformers can not benefit from simply deepening layers after a certain depth. Earlier studies hypothesize that this may be due to issues such as the attention or feature collapse or due to uniformity among patches or tokens [101, 25, 92]. Dong et al. [18] also point out that the output of a pure Transformer, i.e., an attention mechanism without skip connections or MLPs, tends to converge to a rank-1 matrix [18]. This analysis is followed by [53], which suggests that rank collapses incur vanishing gradients of attention queries and keys.

In this context, the self-attention acts as a low-pass filter, since the self-attention calculates the weighted average of the value vectors of tokens. Wang et al. [80, Theorem 1] also reveal that the self-attention is a low-pass filter, continuously reducing high-frequency information. This nature contributes to the oversmoothing phenomenon as unique high-frequency features are lost in deeper layers of the network, further worsening the uniformity of token representations. Therefore, we extend the term "oversmoothing" to describe the degeneration challenge observed in Transformers.

There have been proposed many empirical countermeasures for ViT, such as patch diversification [102, 25], rank collapse alleviation [101, 99], and training stabilization [74, 98]. Similar alleviating methods have been also proposed in the field of NLP, such as unsupervised learning [9], and resolve the oversmoothing and the token uniformity (or information diffusion) problems [18, 92]. There are studies on utilizing high frequency information via frequency domain analyses [80, 4], but they are not designed on top of graph filtering perspectives. Dovonon et al. [20] find that Transformers are not inherently low-pass filters, but oversmoothing depends on the eigenspectrum of the self-attention layers. They propose a reparametrization of the Transformer weights, ensuring that oversmoothinng does not occur.

Our paper addresses the oversmoothing problem with graph filters since the self-attention mechanism is a basic graph filtering operation as seen in the previous subsection.

## 3 Graph Filter-based Self-Attention Layers

Let $\bar{\boldsymbol{A}} \in [0,1]^{n \times n}$, where $n$ is the number of tokens in the input to the self-attention layer, be a self-attention matrix. Since Transformers use multi-head self-attentions, there are multiple such matrices. For simplicity, but without loss of generality, we discuss only one head in one layer.

From the GSP perspective, using $\bar{\boldsymbol{A}}$ as the shift operator, a graph filter can be represented as a matrix polynomial filter, as mentioned in Section 2.2. We aim to design this matrix polynomial filter using the two lowest-order terms and one high-order term in Eq. (3). The following theorem shows that,

despite using the three terms, the filter can be either a low-pass filter or a high-pass filter, depending on the coefficient values.

**Theorem 3.1** (Filter characteristics based on coefficient values). *Let $\bar{\boldsymbol{A}}$ be a self-attention matrix interpreted as a graph with connected components. Consider the polynomial graph filter defined by $\sum_{k=0}^{K} w_k \bar{\boldsymbol{A}}^k$, where $w_2, w_3, \ldots, w_{K-1} = 0$ and only $w_0$, $w_1$, and $w_K$ are non-zero. If the coefficients $w_k$ for $k = 0, 1, K$ are positive and their sum is 1, then the polynomial filter acts as a low-pass filter, attenuating high-frequency components and promoting smoothness across the graph. Conversely, if $w_k = (-\alpha)^k$ for $k = 0, 1, K$ and $\alpha \in (0, 1)$ with sufficient large $K$, the polynomial filter exhibits high-pass filter behavior.*

The proof of Theorem 3.1 is in Appendix G. Based on Theorem 3.1, we propose to use the following graph filter, $\boldsymbol{H}_{\text{GFSA}}$, where the two lowest-order terms and one high-order term of the matrix polynomial are used:

$$\boldsymbol{H}_{\text{GFSA}} = w_0 \boldsymbol{I} + w_1 \bar{\boldsymbol{A}} + w_K \bar{\boldsymbol{A}}^K, \tag{4}$$

where $w_0$, $w_1$, $w_K$ are coefficients and $K$ is a hyper-parameter where $K \geq 2$. The coefficients can be learnable weights and we learn them with gradient descent algorithms.

**Approximation of the high-order term.** In Eq. (4), it is costly to calculate $\bar{\boldsymbol{A}}^K$ when $K$ is large, so we need a way to approximate the high-order term $\bar{\boldsymbol{A}}^K$ in GFSA. We use the first-order Taylor approximation at point $a = 1$ for this purpose:

$$f(x) \simeq f(a) + f'(a)(x - a), \tag{5}$$

thus, we approximate $f(K) = \bar{\boldsymbol{A}}^K$ as follows:

$$f(K) = \bar{\boldsymbol{A}}^K \simeq f(1) + f'(1)(K - 1). \tag{6}$$

Computing the derivative of $\bar{\boldsymbol{A}}^K$ directly at the evaluation point requires high computational costs. To overcome this problem, we adopt the forward finite difference method, which approximates derivatives with the difference term:

$$f'(K) = \frac{f(K + h) - f(K)}{h} = \frac{\boldsymbol{A}^{K+h} - \boldsymbol{A}^K}{h}, \tag{7}$$

where the approximation error is $\mathcal{O}(h^2)$. To balance the trade-off between computational efficiency[2] and accuracy, we set $h = 1$. This method is inspired by the approach in Brouwer et al. [6], which uses the difference term between two consecutive hidden states in discrete time to approximate the derivatives. Therefore, we approximate $\bar{\boldsymbol{A}}^K$ as:

$$\begin{aligned} f(K) = \bar{\boldsymbol{A}}^K &\simeq f(1) + (\bar{\boldsymbol{A}}^2 - \bar{\boldsymbol{A}})(K - 1) \\ &= \bar{\boldsymbol{A}} + (K - 1)(\bar{\boldsymbol{A}}^2 - \bar{\boldsymbol{A}}). \end{aligned} \tag{8}$$

The approximation for $\bar{\boldsymbol{A}}^K$ with $\bar{\boldsymbol{A}}$ and $\bar{\boldsymbol{A}}^2$ provides a simpler computation that can significantly reduce the required computational resources and time.

**GFSA: our graph filter-based self-attention.** Our proposed graph filter-based self-attention (GFSA) is defined with the graph filter $\tilde{\boldsymbol{H}}_{\text{GFSA}}$ as follows:

$$\text{GFSA}(\boldsymbol{X}) := \tilde{\boldsymbol{H}}_{\text{GFSA}} \boldsymbol{X} \boldsymbol{W}_{\text{val}}, \tag{9}$$

$$\tilde{\boldsymbol{H}}_{\text{GFSA}} = w_0 \boldsymbol{I} + w_1 \bar{\boldsymbol{A}} + w_K (\bar{\boldsymbol{A}} + (K - 1)(\bar{\boldsymbol{A}}^2 - \bar{\boldsymbol{A}})), \tag{10}$$

where the last term is the approximated $\bar{\boldsymbol{A}}^K$ from Eq. (8). We replace the original self-attention layer in various Transformers with the proposed graph filter-based layer without changing other parts. Therefore, GFSA can be plugged into any Transformers that rely on the self-attention. For pseudocode, see Appendix I.

---

[2]Calculating the power of a matrix for small $h$ requires a high computational cost since it is calculated by diagonalizing the matrix or using Schur normal form [31].

# 4 Properties of GFSA

This section analyzes the theoretical error of the $\bar{A}^K$ approximation used by GFSA and how GFSA can mitigate oversmoothing. We also explain the meaning of GFSA's high-order term in the context of Transformers and provide comparisons of GFSA in other models.

**Theoretical characteristics of approximation error in GFSA.** We provide a theorem that provides an upper bound on the error introduced by approximating the power of a matrix, specifically using the first-order Taylor expansion. The following theorem specifically analyzes the error of matrix $\bar{A}^K$ when approximated using a first-order Taylor expansion.

**Theorem 4.1** (Error bound for approximated high-order term in GFSA). *Define the error term, $E_K$, as the difference between the exact value and approximated value of $\bar{A}^K$, which is given by $E_K = ||\bar{A}^K - (\bar{A} + (K-1)(\bar{A}^2 - \bar{A}))||_F$, where $|| \cdot ||_F$ denotes the Frobenius norm. Then, the error bound can be shown that $E_K \leq 2\sqrt{n}K$.*

The error bound provides an upper limit for the difference between the actual value of $\bar{A}^K$ and its approximation. The proof of Theorem 4.1 is in Appendix H. It shows theoretical validity for using approximations to the high-order term in the filters of our GFSA. In terms of performance, we report a difference of approximately $\bar{A}^K$ between the actual calculated values in Appendix J.

**How to alleviate the oversmoothing problem?** The key leading to the low/high pass filtering behavior of our proposed filter is the coefficients $\{w_0, w_1, w_K\}$ — note that in the self-attention of Transformers, $w_0 = w_K = 0$ and $w_1 = 1$. Since our method can learn any appropriate values for them for a downstream task, it can be reduced to low-pass-only, high-pass-only, or combined filters. According to Theorem 3.1, our graph polynomial filter can be said to be a low-pass filter when $w_1, w_K$ are positive and a high-pass filter when they are negative. Therefore, our method can learn the appropriate coefficients $\{w_0, w_1, w_K\}$ for downstream tasks, so it can be reduced to a low-pass-only, high-pass-only, or combined filter, alleviating the oversmoothing problem of self-attention.

**The meaning of the high-order term in GFSA in the context of Transformers.** Existing self-attention only captures simple pairwise similarities between tokens and is limited in capturing high-order dependencies. For example, given the two sentences, "Books are more expensive than pencils" and "Books are cheaper than computers", to understand the relationship between "computers" and "pencils", we need to capture the high-order dependencies connected through the "Book" token. However, it is difficult to capture these high-order dependencies with traditional self-attention [103]. Therefore, from a Transformer perspective, the approximated $\bar{A}^K$ in GFSA can be interpreted as being able to capture these high-order dependencies.

**Comparison to Transformers.** In the field of computer vision, there has been recent research on adjusting the frequency response of ViT. HAT [4] creates adversarial examples by altering clean images with high-frequency perturbations and jointly trains the ViT on clean images and adversarial examples. Through this, they aim to solve the problem of the ViT being unable to capture high-frequency by allowing us to capture the high-frequency components of the images. However, HAT has the disadvantage of requiring more epochs than the existing ViT, as it must perform adversarial training in some initial epochs and train normally in the remaining epochs. Wang et al. [80] use the concept of DSP, which is a special case of GSP, to isolate the lowest frequency component in the Fourier domain and use a filter learned by rescaling the low and high-frequency components. On the other hand, our GFSA extends the concept to graph signal processing and redesigns self-attention as a graph filter. While GFSA seeks to design a better graph filter by interpreting self-attention as a graph filter, Shi et al. [71] are inspired by JKNet [91], and they solve the oversmoothing problem by fusing the hidden vectors of each layer. However, their method has a limitation with memory increasing, and they only applied it to BERT.

**Comparison to GCNs.** Comparisons to GCNs that can be interpreted as graph filters [39, 15, 24] are inevitable. GFSA without a high-order term is analogous to ChebNet [15] with $K = 1$. In addition, GFSA reduces to the vanilla GCN [39] when $K = 1$, $w_0 = 0$, $w_1 = 1$. GPR-GNN [12], which approximates graph convolutions using the monomial basis, is identical to GFSA if it only considers up to first order and additionally uses a $K$-order term and learns the coefficients. When we

use only a high-order term and $w_K$ is learned to a negative value, GFSA can become similar to the reaction-diffusion layer of GREAD [13], $\bar{A}X + \beta(\bar{A} - \bar{A}^2)$, depending on the higher order terms.

## 5 Experiments

In this section, we demonstrate the effectiveness of GFSA through a series of experiments. These experiments encompass various tasks: i) language understanding and causal language modeling, ii) image classification, iii) graph-level task, and iv) code classification. We replace the self-attention of base Transformers in those fields with our GFSA. Our modification adds only tens to hundreds of parameters, which are negligible in comparison with the original size of base models.

### 5.1 Experiments on Natural Language Understanding

**Setting.** We integrate GFSA into 3 pre-trained large language models: BERT, ALBERT, and RoBERTa. We evaluate them on the GLUE benchmark, which includes 3 categories of natural language understanding tasks: i) single-sentence, ii) similarity and paraphrasing, and iii) natural language inference tasks. For each task, we select the best hyperparameters for GFSA, and the other hyperparameters are fixed. The detailed experimental settings are in Appendix K.1.

**Results.** The results are shown in Table 1. When GFSA was plugged into backbones, average performance scores improved across all models over pure backbones. This indicates that GFSA is effective in both large models like BERT and RoBERTa, as well as relatively smaller models like ALBERT. It is worth mentioning that in the case of RoBERTa finetuned on the CoLA dataset; there is a significant margin increase from 60.34% to 64.11%, which is a 3.77% improvement with only 144 additional parameters. When compared to ContraNorm, GFSA shows a greater performance improvement on average. Fig. 5 in Appendix C shows that these performance enhancements can be attributed to addressing the oversmoothing issue through the designed graph filter.

Table 1: Results comparison on GLUE benchmark. **Avg** denotes the average performance.

| Method | #Params | CoLA | SST-2 | MRPC | QQP | STS-B | MNLI-m/mm | QNLI | RTE | **Avg** |
|---|---|---|---|---|---|---|---|---|---|---|
| BERT$_{\text{BASE}}$ [16] | 110M | 56.79 | 93.81 | 88.70 | 88.32 | 88.16 | 84.96/84.15 | 91.63 | 66.06 | 82.51 |
| + ContraNorm | 110M | **59.89** | 93.92 | 89.88 | **88.51** | **88.36** | 85.11/84.50 | 91.84 | **69.31** | 83.48 |
| + GFSA | 110M | 59.56 | **94.15** | **90.60** | 88.46 | 88.33 | **85.12/85.06** | 91.95 | 68.95 | **83.58** |
| ALBERT$_{\text{BASE}}$ [40] | 11M | 57.86 | 92.32 | 91.80 | 85.30 | 90.37 | **85.37**/84.37 | 91.76 | 76.90 | 84.01 |
| + ContraNorm | 11M | 57.45 | 93.00 | **92.83** | 87.78 | 90.55 | 85.06/84.57 | **92.28** | **78.70** | 84.69 |
| + GFSA | 11M | **60.21** | **93.23** | 92.47 | **87.79** | **90.63** | 85.29/**84.92** | 92.17 | **78.70** | **85.05** |
| RoBERTa$_{\text{BASE}}$ [44] | 125M | 60.34 | 94.84 | 92.28 | 88.86 | 89.99 | 87.94/87.30 | 92.57 | 78.70 | 85.87 |
| + ContraNorm | 125M | 63.06 | **95.41** | 93.17 | 88.91 | 90.34 | 87.88/87.40 | 92.82 | **80.51** | 86.61 |
| + GFSA | 125M | **64.11** | **95.41** | **93.52** | **89.09** | **90.35** | **87.99/87.54** | **92.97** | 80.14 | **86.79** |

### 5.2 Experiments on Causal Language Modeling

**Setting.** We also validate the effectiveness of GFSA on causal language modeling problems. We finetune GPT2 [61] on the following 3 datasets: Penn Treebank (PTB) [47], WikiText-2, and WikiText-103 [50]. Following the evaluation method in Yao et al. [93], we finetune models for 15 epochs with PTB, 4 epochs with WikiText-103, and 10 epochs with WikiText-2, and report the perplexity for sensitivity metric. The detailed experimental settings are in Appendix L.1.

**Results.** Table 2 shows the perplexity on PTB, WitiText-2, and WikiText-103. Across all datasets, GPT2 with GFSA consistently outperforms the vanilla GPT2. Our GFSA improves the average perplexity from 18.806 to 18.764. Note that performance improvements are made with only 144 additional learnable parameters for 12 layers with 12 heads.

Table 2: Results comparison on GPT-2 finetuned with GFSA. **Avg** denotes the average performance.

| Method | #Params | PTB | WikiText-2 | WikiText-103 | **Avg** |
|---|---|---|---|---|---|
| GPT2 | 117M | 19.513 | 20.966 | 15.939 | 18.806 |
| + GFSA | 117M | **19.450** | **20.923** | **15.919** | **18.764** |

## 5.3 Experiments on Vision Transformers

**Setting.** We aim to demonstrate the efficacy of our GFSA across a spectrum of ViT backbones. We choose DeiT [74], CaiT [75], and Swin [45] as the backbone, and the models are trained from scratch. When training the 12-layer DeiT, we follow the same training recipe, hyperparameters, and data augmentation from Touvron et al. [74]. For detailed experimental settings, see Appendix M.1.

**Results.** The experimental evaluations are summarized in Table 3. We compare various models on the ImageNet-1k benchmark. The results show that the proposed GFSA successfully enhances DeiT, CaiT, and Swin across all depth settings and training methods. GFSA provides additional parameters less than 72 for 12-layer DeiT while improving top-1 accuracy by 1.63%. To sum up, we observed that both shallow and deep ViTs can achieve the following benefits from GFSA: i) The filter response shows GFSA can preserve higher-frequency representation (cf. Fig. 2 (a)) and ii) Fig. 2 (b) shows that GFSA mitigates the increase in the cosine similarity of representation as the layer gets deeper. We further compare with state-of-the-art models that use Fourier transforms rather than graph filters in Appendix M.5. We also show results under the same settings as ContraNorm [28] in Appendix M.6.

Table 3: Results comparison on ImageNet-1k. Our full results with other models are in Appendix M.4.

| Category | Method | Input Size | #Layers | #Params | Top-1 Acc |
|---|---|---|---|---|---|
| Transformer | DeiT-S [74] | 224 | 12 | 22M | 79.8 |
| | DeiT-S + AttnScale [80] | 224 | 12 | 22M | 80.7 |
| | DeiT-S + FeatScale [80] | 224 | 12 | 22M | 80.9 |
| | DeiT-S + ContraNorm [28] | 224 | 12 | 22M | 80.4 |
| | Swin-S [45] | 224 | 12 | 50M | 82.9 |
| | DeiT-S [74] | 224 | 24 | 43M | 80.5 |
| | CaiT-S [75] | 224 | 24 | 47M | 82.6 |
| | DeiT-S + AttnScale [80] | 224 | 24 | 44M | 81.1 |
| | DeiT-S + FeatScale [80] | 224 | 24 | 44M | 81.3 |
| | DeiT-S + ContraNorm [28] | 224 | 24 | 43M | 80.7 |
| GFSA | DeiT-S + GFSA | 224 | 12 | 22M | **81.1** |
| | DeiT-S + GFSA | 224 | 24 | 43M | **81.5** |
| | CaiT-S + GFSA | 224 | 24 | 47M | **82.8** |
| | Swin-S + GFSA | 224 | 12 | 50M | **83.0** |

## 5.4 Experiments on Graph-level Tasks

**Setting.** To evaluate the efficacy of GFSA on graph-level tasks, we conduct experiments on a broader range of datasets. We use datasets from Long-Range Graph Benchmark (LRGB) [21] (e.g., Peptide-func and Peptide-struct), Benchmarking GNNs [22] (e.g., ZINC, MNIST, CIFAR10), Open Graph Benchmark (OGB) dataset [32] (e.g., Molhiv and MolTox21), and OGB-LSC dataset (i.e., PCQM4M-LSC) [33]. We choose Graphormer [94], Graph-ViT [30], and GPS [63] as our backbone architectures, following their original experimental protocols for fair comparison. For GPS, we replace its self-attention module with our GFSA while maintaining its best configuration and other hyperparameters. For Graph-ViT, we apply GFSA to the Hadamard self-attention method, which He et al. [30] propose as optimal. For a detailed experimental setting, see Appendix O.1.

**Results.** Tables 4, 5, and 6 show consistent performance improvements when GFSA is integrated with backbone architectures. Graph-ViT + GFSA shows improvements on all datasets. On Peptide-func, it achieves a 0.65% increase in AP. Notably, in PCQM4M, incorporating GFSA improves the validation MAE by 7.20%. Due to space constraints, the results with standard deviation are included in Appendix O.2.

## 5.5 Experiments on Automatic Speech Recognition

**Setting.** We conduct automatic speech recognition (ASR) experiments on the LibriSpeech [3] dataset [55], which consists of audio recordings paired with their transcriptions. We use Branch-

---

[3] http://www.openslr.org/12

Table 4: Results on ZINC

| Method | #Params | MAE ($\downarrow$) |
|---|---|---|
| Graphormer | 500K | 0.1240 |
| + GFSA | 500K | **0.1189** |

Table 5: Results on PCQM4M and PCQM4Mv2

| Method | #Params | PCQM4M | | PCQM4Mv2 | |
|---|---|---|---|---|---|
| | | Train ($\downarrow$) | Validate ($\downarrow$) | Train ($\downarrow$) | Validate ($\downarrow$) |
| Graphormer | 48.3M | 0.0535 | 0.1286 | 0.0250 | 0.0862 |
| + GFSA | 48.3M | **0.0312** | **0.1193** | **0.0249** | **0.0860** |

Table 6: Experimental evalutation of GFSA plugged into GPS and Graph-ViT. Results marked with † indicate settings where we conducted our own experiments due to unavailable Hadamard self-attention performance in He et al. [30]'s paper.

| Method | Peptide-func | Peptide-struct | MNIST | CIFAR10 | Molhiv | MolTOX21 | ZINC |
|---|---|---|---|---|---|---|---|
| | AP ($\uparrow$) | MAE ($\downarrow$) | Accuracy ($\uparrow$) | Accuracy ($\uparrow$) | ROCAUC ($\uparrow$) | ROCAUC ($\uparrow$) | MAE ($\downarrow$) |
| GPS | $0.6535_{\pm0.0041}$ | $0.2500_{\pm0.0005}$ | $0.9805_{\pm0.0013}$ | $0.7230_{\pm0.0036}$ | – | – | $0.070_{\pm0.004}$ |
| + GFSA | $\mathbf{0.6593}_{\pm0.0094}$ | $\mathbf{0.2496}_{\pm0.0013}$ | $\mathbf{0.9814}_{\pm0.0014}$ | $\mathbf{0.7244}_{\pm0.0048}$ | – | – | $\mathbf{0.069}_{\pm0.002}$ |
| Graph-ViT | $0.6919_{\pm0.0085}$ | $^{\dagger}0.2474_{\pm0.0016}$ | $0.9820_{\pm0.0005}$ | $0.6967_{\pm0.0040}$ | $0.7792_{\pm0.0149}$ | $0.7851_{\pm0.0077}$ | $0.0849_{\pm0.0047}$ |
| + GFSA | $\mathbf{0.6964}_{\pm0.0025}$ | $\mathbf{0.2461}_{\pm0.0024}$ | $\mathbf{0.9826}_{\pm0.0004}$ | $\mathbf{0.6987}_{\pm0.0028}$ | $\mathbf{0.7830}_{\pm0.0109}$ | $\mathbf{0.7895}_{\pm0.0069}$ | $\mathbf{0.0845}_{\pm0.0032}$ |

former [59] and a pure Transformer. For implementation, we follow the recipes of SpeechBrain [65] and the detailed settings are in Appendix N.1.

**Results.** Table 7 compares word error rates (WERs) on LibriSpeech 100h and 960h. For 100h, Transformer+GFSA achieves 10.30/25.30 on the test clean/other set, which is a 6.53% improvement over the Transformer for the WER of the test clean. For 960h, Transformer+GFSA shows a WER result of 2.31 in test clean, a 4.55% improvement over Transformer and Branchformer+GFSA achieves 2.31/5.49 with an LM on the test clean/other sets. Fig. 8 in Appendix N.2 depicts the learning curves of train loss and valid loss when using GFSA, showing the effectiveness of our proposed filter.

Table 7: Results for ASR training on LibriSpeech 100h and 960h with GFSA

| Method | #Params | LibriSpeech 100h | | LibriSpeech 960h | |
|---|---|---|---|---|---|
| | | test-clean WER | test-other WER | test-clean WER | test-other WER |
| Transformer | 71.5M | 11.02 | 25.42 | 2.42 | 5.50 |
| + GFSA | 71.5M | **10.30** | **24.30** | **2.31** | **5.49** |
| Branchformer | 109.8M | 9.63 | 22.43 | 2.13 | 5.00 |
| + GFSA | 109.8M | **9.60** | **22.25** | **2.11** | **4.94** |

## 5.6 Experiments on Code Classification

**Setting.** We conduct a code defect detection task based on Devign dataset provided by Zhou et al. [105]. We use RoBERTa [44], CodeBERT [23], PLBART [2], and CodeT5 [84] as our backbone models. The detailed settings are in Appendix P.1.

**Results.** Table 8 shows the accuracy of all models; GFSA results better than the base models. The biggest improvement is 2.40% for RoBERTa. In the case of CodeT5-base, using GFSA shows an accuracy of 64.75, an improvement of 1.95% from 63.51. CodeT5-small+GFSA has only about 100 additional parameters compared to CodeT5-small with 60M parameters, and even more impressively, it surpasses the accuracy of CodeT5-base. The biggest improvement is 2.40% for RoBERTa. In Appendix P.2, we include case studies for this task. We also report the results of the code clone detection task in Appendix Q.

Table 8: Results on code classification. The number in ($\uparrow$) indicates the improvement rate.

| Method | Accuracy |
|---|---|
| RoBERTa | 62.88 |
| + GFSA | **64.39** ($\uparrow$ 2.40%) |
| CodeBERT | 64.31 |
| + GFSA | **64.49** ($\uparrow$ 0.12%) |
| PLBART | 62.63 |
| + GFSA | **62.96** ($\uparrow$ 0.52%) |
| CodeT5-small | 63.25 |
| + GFSA | **63.69** ($\uparrow$ 0.70%) |
| CodeT5-base | 63.51 |
| + GFSA | **64.75** ($\uparrow$ 1.95%) |

# 6 Discussion on Runtime Overheads

**Limitation.** The introduction of our GFSA layer results in a slight increase in training and inference time. We report the runtimes when plugging GFSA in Appendices R and S. For GLUE benchmark, integrating GFSA into BERT enhances the average performance from 82.51% to 83.58% (see Table 1) with more overhead of less than 36 seconds per epoch based on average training time (see Table 23). Considering the improvements, the increases in training time are negligible.

**GFSA in selected layers: a strategy to mitigate the limitation.** As GFSA requires more calculation than the original self-attention, the runtime after using GFSA slightly increases. Our experiments initially applied GFSA across all Transformer layers (as discussed in Section 5); however, to reduce computational load, we propose a selective application strategy. For this purpose, GFSA is used only on even-numbered layers. In Tables 35 to 39 of Appendix T, the results show that this strategy effectively reduces runtime increases while preserving comparable performance to the full-layer GFSA integration. Notably, the selective application of GFSA cuts the per-epoch runtime increase by 26.90% relative to its full-layer application, with only a 7.39% increase in runtime per epoch compared to the backbone model in Table 39.

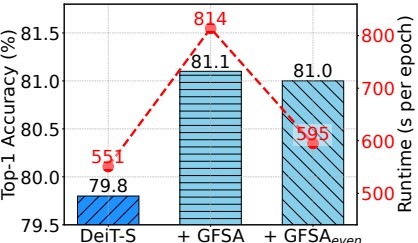

Figure 3: Effectiveness of our selective layer strategy on ImageNet-1k. This shows out strategy's ability to maintain accuracy benefits while mitigating runtime increases.

**GFSA in linear Transformers.** Although GFSA requires additional computation for calculating $\bar{A}^2$, we explore integrating GFSA with linear attention variants to maintatin efficiency and scalability. Recent approaches [36, 70] achieve linear complexity by reformulating softmax operations and reordering matrix multiplication in self-attention. We apply similar principles to compute second-order self-attention efficiently, enabling $\tilde{H}_{\text{GFSA}}$ calculation with linear complexity with respect to sequence length. Fig. 4 shows the performance, runtime and GPU usage changes when applying our GFSA to Transformers with linear complexity. GFSA still improves performance compared to the backbone model, while the increase in time and GPU usage is minimal. Notably, when GFSA is applied to Efficient Attention [70], the performance is improved and the runtime is 11.82 times faster

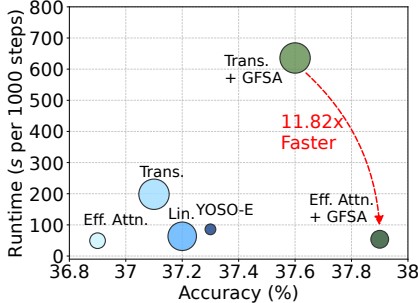

Figure 4: Performance ($x$-axis), runtime ($y$-axis), and GPU usage (circle sizes) of various Transformers and integrated GFSA on Long-Range benchmark

than when GFSA is applied to the vanilla self-attention. This shows that GFSA can be effectively implemented with linear complexity architectures while preserving its benefits and providing a solution for addressing computational concerns.

# 7 Conclusion

Our proposed GFSA achieves high performance with improvements on a variety of tasks. GFSA is a simple yet effective method that enriches self-attention in Transformers with more diverse frequency information. This enables GFSA to address the oversmoothing problem and learn better latent representations for downstream tasks. However, our GFSA does not bring significant overheads in those Transformers' empirical runtime complexity. One can use more complicated graph filters to enhance accuracy more, but our goal is to find a balance between accuracy enhancements and overheads in runtime complexity.

We believe that GFSA suggests a promising new direction for improving Transformers. GFSA can be implement with simple way and used in conjunction with other techniques to further improve the performance of Transformers. Considering the ongoing advancements in large language models, such as GPT-4 [1] and LLaMA [76], we hope that our approach may offer new insights for enhancing their performance and efficiency.

## Acknowledgements

N. Park was partly supported by the Korea Advanced Institute of Science and Technology (KAIST) grant funded by the Korea government (MSIT) (No. G04240001, Physics-inspired Deep Learning, 10%), Institute for Information & Communications Technology Planning & Evaluation (IITP) grants funded by the Korea government (MSIT) (No. RS-2020-II201361, Artificial Intelligence Graduate School Program (Yonsei University), 20%; No. RS-2024-00457882, AI Research Hub Project, 50%), and Samsung Electronics Co., Ltd. (No. G01240136, KAIST Semiconductor Research Fund (2nd), 10%). K. Lee acknowledges support from the U.S. National Science Foundation under grant IIS 2338909. Dr. Trask acknowledges funding under the Department of Energy under the Mathematical Multifaceted Integrated Capability Centers program.

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

# Appendix

## A  Reproducibility Statement

To ensure the reproducibility and completeness of this paper, we include the Appendix with 12 sections. Appendix I provides our PyTorch-style pseudo code for our GFSA method. The pseudo code helps to implement our GFSA to any Transformers used a pure self-attention. All experiments in the paper are reproducible with additional implementation details provided in Appendices K to Q.

## B  Broader Impact

In terms of the broader impact of this research on society, we do not see the very negative impacts that might be expected. However, this paper may have implications for the carbon footprint and accessibility of learning algorithms. The computations required for machine learning research are rapidly growing, resulting in a larger carbon footprint [69]. Our study improves performance and increases runtime very slightly, but the runtime increase is not very significant. However, in future research, it will also be important to study and improve our GFSA by taking carbon footprints into account.

GFSA improves the performance of existing Transformer-based models, which can have many positive impacts on society through services that utilize natural language processing, computer vision, and speech recognition. However, it will also be important to improve GFSA by considering other dimensions of AI, such as robustness to adversarial examples, fairness, and explainability.

## C  Oversmoothing and Additional Visualizations

In Fig. 2, we show the visualizations of oversmoothing characteristics in DeiT. We also provide visualizations in other domains. We show the filter response, cosine similarity, and singular value of BERT finetuned on STS-B dataset of GLUE tasks in Fig. 5 and Graphormer finetuned on ZINC dataset in Fig. 6.

To characterize self-attention, we first analyze the filter response of self-attention in the frequency domain. We follow the method used by Wang et al. [80] for spectral visualization of the self-attention matrix. As shown in Fig. 2 (a), DeiT has a near-zero magnitude for the high frequencies, which is characteristic of a low-frequency filter and is likely to result in oversmoothing when applied multiple times.

We follow the calculation method of Guo et al. [28] for cosine similarity. As shown in Fig. 2 (b), the higher similarity as the layers of the model get deeper is related to the oversmoothing problem. To further analyze this issue, we also consider the dimensionality collapse in Transformer-based models. We plot the singular value distribution of the feature in the last block. As shown in Fig. 2 (c), insignificant, near-zero values dominate the feature distribution. As layers get deeper, the similarity of features increases and dimensional collapse occurs. The oversmoothing problem is the same in BERT and Graphormer, as shown in Fig. 5 and Fig. 6.

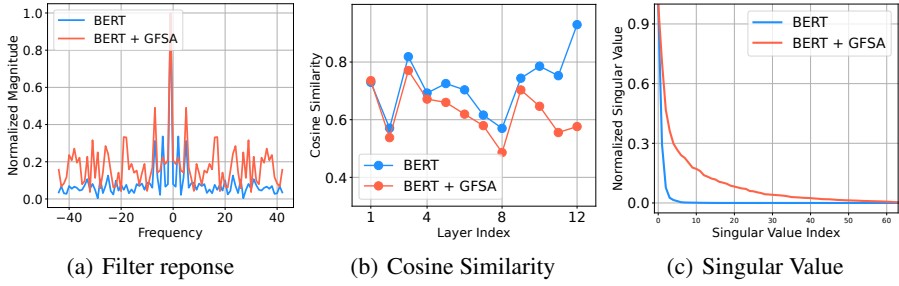

(a) Filter reponse       (b) Cosine Similarity       (c) Singular Value

Figure 5: Filter frequency response, cosine similarity, and singular values on STS-B for BERT and BERT+GFSA

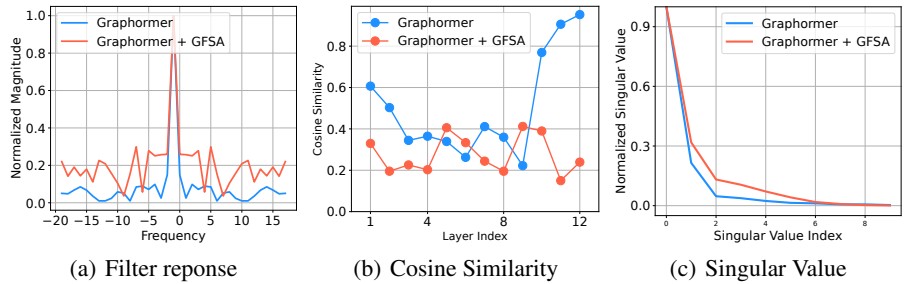

(a) Filter reponse      (b) Cosine Similarity      (c) Singular Value

Figure 6: Filter frequency response, cosine similarity, and singular values on ZINC for Graphormer and Graphormer+GFSA

# D Analysis of Frequency Responses with Visualizations

We analyze the frequency responses, which represent the impact of learned coefficients, for all 12 layers of BERT$_{BASE}$ with and without GFSA. From Fig. 7, our analysis reveals that GFSA learns various filter types between layers. In early layers, we observe a tendency towards low-pass filtering, with prominent peaks at low frequencies. This aligns with the need for broader feature extraction in initial layers. The middle layers show a mix of low-pass and high-pass characteristics, with more complex frequency responses. This suggests GFSA is learning to balance between feature extraction and refinement. In deep layers, there is a noticeable shift towards higher frequency responses, indicating a move towards high-pass filtering. This shift supports our claim that GFSA can mitigate oversmoothing in deeper layers. BERT$_{BASE}$+GFSA shows a consistently higher magnitude response at higher frequencies, especially in deeper layers, compared to vanilla BERT. In other word, vanilla self-attention works primarily as a low-pass filter, while GFSA utilizes a wider range of frequencies.

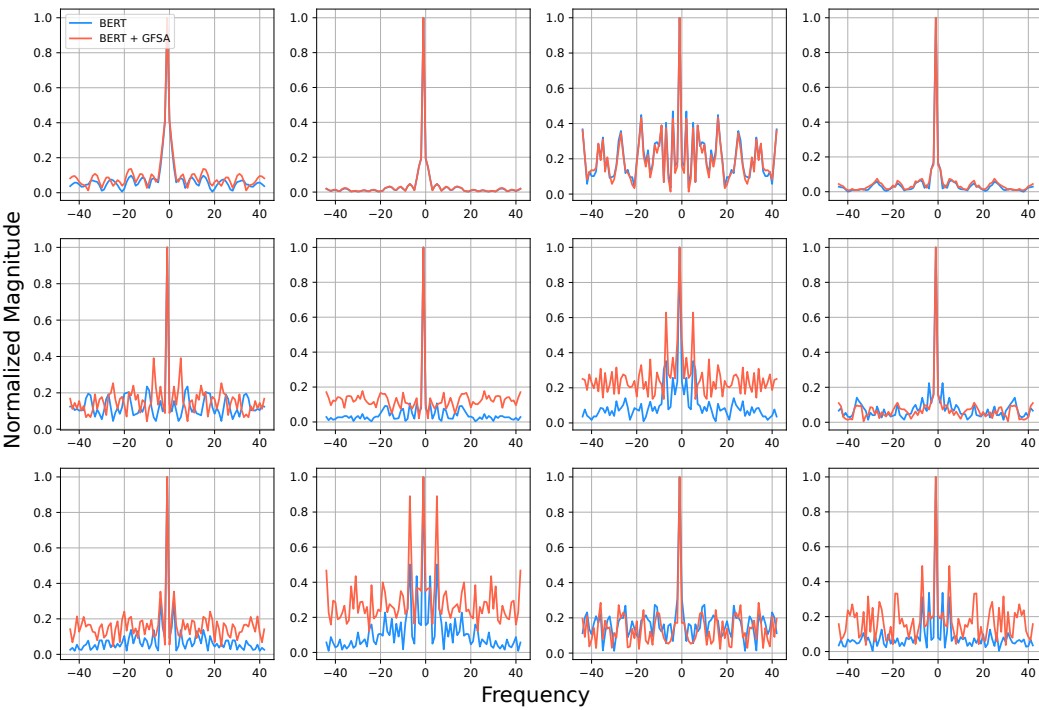

Figure 7: Visualization of the frequency responses for all 12 layers of BERT trained on STS-B dataset. The top-left figure corresponds to the first layer, and the bottom-right figure corresponds to the last layer.

## E  Frequency Analyses in the Singular Value Domain

Graph signal processing (GSP) [67, 68] can be understood as a generalized concept of DSP — in other words, DSP is a special case of GSP where a *line graph with n nodes* is used and therefore, the graph Fourier transform (GFT) of the line graph is identical to the discrete Fourier transform.

In the definition of GFT, we assume that the graph shift operator (GSO) $\boldsymbol{S}$ is diagonalizable. Considering the eigendecomposition of the GSO $\boldsymbol{S} = \boldsymbol{V}^{\mathsf{T}}\boldsymbol{\Lambda}\boldsymbol{V}$ with eigenvector $\boldsymbol{V}$, we can write the graph filter output as follows:

$$\boldsymbol{y} = \sum_{k=0}^{K} w_k \boldsymbol{S}^k \boldsymbol{x} = \sum_{k=0}^{K} \boldsymbol{V}^{\mathsf{T}} w_k \boldsymbol{\Lambda}^k \boldsymbol{V} \boldsymbol{x} = \boldsymbol{V}^{\mathsf{T}} \Big( \sum_{k=0}^{K} w_k \boldsymbol{\Lambda}^k \Big) \boldsymbol{V} \boldsymbol{x} = \boldsymbol{V}^{\mathsf{T}} g(\boldsymbol{\Lambda}) \boldsymbol{V} \boldsymbol{x}, \qquad (11)$$

where $\boldsymbol{x} \in \mathbb{R}^n$ is a 1-dimensional graph signal, $\boldsymbol{\Lambda}$ is a diagonal matrix with eigenvalues, and $w_k \in [-\infty, \infty]$ is a coefficient.

However, one can use the singular value decomposition, when the GSO is not diagonalizable but symmetrically normalized, instead of the eigendecomposition [49]. Both the singular value decomposition and the eigendecomposition project the original signal onto a set of basis, but they use different basis sets. In the singular value decomposition, we sort the set of basis in ascending order of their eigenvalues, and perform frequency domain-like analyses [106, 49].

Since the self-attention matrix's row-wise sum is always 1, the following is the case: $\bar{\boldsymbol{A}} = \boldsymbol{D}^{-1}\boldsymbol{A} = \frac{1}{n}\boldsymbol{A}$, where $n$ is the number of tokens. Maskey et al. [49] define the following symmetrically normalized adjacency matrix (SNA): $\boldsymbol{D}_{in}^{-1/2}\boldsymbol{A}\boldsymbol{D}_{out}^{-1/2}$. Since the degree of every node is $n$ in the self-attention matrix, the following is the case: $\boldsymbol{D}_{in}^{-1/2}\boldsymbol{A}\boldsymbol{D}_{out}^{-1/2} = \boldsymbol{D}^{-1/2}\boldsymbol{A}\boldsymbol{D}^{-1/2} = \frac{1}{\sqrt{n}}\boldsymbol{A}\frac{1}{\sqrt{n}} = \frac{1}{n}\boldsymbol{A} = \bar{\boldsymbol{A}}$. Therefore, the self-attention matrix is a special case of SNAs.

## F  Matrix Polynomial vs. Graph Fourier Transform

There are two paradigms of implementing graph filters: i) matrix polynomial, which does not require diagonalizability, and ii) graph Fourier transform, which uses the eigendecomposition for diagonalizable adjacency matrices or uses the Jordan decomposition or the singular value decomposition for non-diagonalizable adjacency matrices.

Those two paradigms have their own weaknesses: i) the matrix polynomial approach requires explicit matrix multiplications, and ii) the graph Fourier transform approach requires expansive spectral decompositions. The matrix polynomial is preferred when there are not many matrix multiplications. Otherwise, the graph Fourier transform approach may be better since the matrix multiplication can be simplified after the decomposition.

Among those two, we use the first matrix polynomial approach with only three non-zero coefficients $\{w_0, w_1, w_K\}$ since it does not require the complicated spectral decomposition. Since we do not rely on any explicit spectral decomposition but on the matrix polynomial, any adjacency matrix can be used.

## G  Proof of Theorem 3.1

**Theorem 3.1** (Filter characteristics based on coefficient values)**.** *Let $\bar{\boldsymbol{A}}$ be a self-attention matrix interpreted as a graph with connected components. Consider the polynomial graph filter defined by $\sum_{k=0}^{K} w_k \bar{\boldsymbol{A}}^k$, where $w_2, w_3, \ldots, w_{K-1} = 0$ and only $w_0$, $w_1$, and $w_K$ are non-zero. If the coefficients $w_k$ for $k = 0, 1, K$ are positive and their sum is 1, then the polynomial filter acts as a low-pass filter, attenuating high-frequency components and promoting smoothness across the graph. Conversely, if $w_k = (-\alpha)^k$ for $k = 0, 1, K$ and $\alpha \in (0, 1)$ with sufficient large $K$, the polynomial filter exhibits high-pass filter behavior.*

Note that without filtering, the singular value ratio is $\left|\sigma_i^0\right|/\left|\sigma_1^0\right| = 1$. In the case where $|g(\sigma_i)/g(\sigma_1)| < 1 \ \forall i \geq 2$, it implies that after applying the graph filter $g$, the lowest frequency component further dominates, indicating that the graph filter acts as a low-pass filter. Conversely, in

the case where $|g(\sigma_i)/g(\sigma_1)| > 1 \; \forall i \geq 2$, it implies that after applying the graph filter $g$, the lowest frequency component $\sigma_i$ no longer dominates, indicating that the graph filter acts as a high-pass filter.

*Proof.* We prove the low-pass filter result. For the case where $w_0$, $w_1$, and $w_K$ are positive and their sum is 1, we show that

$$|g(\sigma_1)| = |w_0 + w_1 + w_K| = 1 \tag{12}$$

Hence, proving Theorem G is equivalent to show $|g(\sigma_i)| < 1$.

$$|g(\sigma_i)| = \left|w_0 + w_1\sigma_i + w_K(\sigma_i + (K-1)(\sigma_i^2 - \sigma_i))\right| < |w_0 + w_1\sigma_i + w_K\sigma_i| = \sigma_i < 1 \tag{13}$$

since $\sigma_i + (K-1)(\sigma_i^2 - \sigma_i) = \sigma_i((K-1)\sigma_i - (K-2)) < \sigma_i((K-1) - (K-2)) = \sigma_i$

For the high-pass filter result, when $w_k = (-\alpha)^k/(k+1)$ where $k = 0, 1, K$ and $\alpha \in (0,1)$, then we show that when $w_0 = 1, w_1 = -\alpha/2$,

$$\left|\frac{\lim_{K\to\infty} g(\sigma_i)}{\lim_{K\to\infty} g(\sigma_1)}\right| = \left|\frac{\lim_{K\to\infty} w_0 + w_1\sigma_i + w_K(\sigma_i + (K-1)(\sigma_i^2 - \sigma_i))}{\lim_{K\to\infty} w_0 + w_1 + w_K(1 + (K-1)(1-1))}\right| \tag{14}$$

$$= \left|\frac{\lim_{K\to\infty} 1 - \frac{\alpha}{2}\sigma_i + \frac{(-\alpha)^K}{(K+1)}(\sigma_i + (K-1)(\sigma_i^2 - \sigma_i))}{\lim_{K\to\infty} 1 - \frac{\alpha}{2} + \frac{(-\alpha)^K}{(K+1)}}\right| \tag{15}$$

$$= \left|\frac{1 - \frac{\alpha}{2}\sigma_i + (\lim_{K\to\infty}\frac{(-\alpha)^K}{(K+1)}(\sigma_i + (K-1)(\sigma_i^2 - \sigma_i)))}{1 - \frac{\alpha}{2}}\right| \tag{16}$$

$$= \left|\frac{1 - \frac{\alpha}{2}\sigma_i}{1 - \frac{\alpha}{2}}\right| > 1 \tag{17}$$

since

$$\lim_{K\to\infty} \frac{(-\alpha)^K}{(K+1)}(\sigma_i + (K-1)(\sigma_i^2 - \sigma_i)) = \lim_{K\to\infty} \frac{(-\alpha)^K}{(K+1)}(K-1)(\sigma_i^2 - \sigma_i) \tag{18}$$

$$= \lim_{K\to\infty} (-\alpha)^K \frac{(K+1)}{(K-1)}(\sigma_i^2 - \sigma_i) = 0 \tag{19}$$

It shows that the graph filter with $w_k = (-\alpha)^k/(k+1)$ for $k = 1, 2, K$ emphasizes high-frequency components and acts as a high-pass filter.

This proof supports that the behavior of the polynomial filter as either a low-pass or high-pass filter directly depends on the sign and values of the coefficients, as specified in Theorem 3.1. $\square$

## H   Proof of Theorem 4.1

*Proof.* The Frobeinus norm of the self-attention is directly related to how far the softmax probabilities are from being uniform. For any matrix $M \in \mathbb{R}^{m \times n}$, we have

$$\|\text{softmax}(M)\|_F = \sqrt{\frac{m + \sum_{i=1}^{m} d_{\chi^2}(S_i, U_n)}{n}}, \tag{20}$$

where $S_i$ is the $i$-th row of softmax$(M)$, $U_n$ is the uniform distribution over $n$ elements, and $d_{\chi^2}(p, q) = \sum_i q_i(p_i/q_i - 1)^2$ is the $\chi^2$-divergence between $p$ and $q$ [14]. The Frobenius norm is maximized when the whole mass of the probabilities is on one element, which is a case for $d_{\chi^2}(S_i, U_n) = n - 1$ and $\|\text{softmax}(M)\|_F = \sqrt{m}$. Therefore, we can calculate the upper bound of Frobenius norm for $\bar{A}$ as follows:

$$\|\bar{A}\|_F \leq \sqrt{n}. \tag{21}$$

Note that $\bar{A} \in \mathbb{R}^{n \times n}$ is a right-stochastic matrix normalized with row-wise softmax: i) all the elements of $\bar{A}$ lie within [0, 1], and ii) the row-sum in $\bar{A}$ is equal to 1. Since the self-attention matrix

is a right-stochastic matrix, the power of the self-attention is also a right-stochastic matrix. Therefore, Eq. (21) is also hold for $\bar{\boldsymbol{A}}^K$ as follows:

$$\|\bar{\boldsymbol{A}}^K\|_F = \sqrt{\sum_{i,j} \bar{\boldsymbol{A}}_{i,j}^K} \leq \sqrt{\sum_{i,j} \bar{\boldsymbol{A}}_{i,j}} = \|\bar{\boldsymbol{A}}\|_F \leq \sqrt{n}. \tag{22}$$

Now, considering the error term $E_K$ as given by Theorem 4.1, and applying the triangle inequality for matrix norms:

$$E_K = \|\bar{\boldsymbol{A}}^K - (\bar{\boldsymbol{A}} + (K-1)(\bar{\boldsymbol{A}}^2 - \bar{\boldsymbol{A}}))\|_F \tag{23}$$
$$\leq \|\bar{\boldsymbol{A}}^K\|_F + \|\bar{\boldsymbol{A}}\|_F + (K-1)(\|\bar{\boldsymbol{A}}^2\|_F + \|\bar{\boldsymbol{A}}\|_F) \tag{24}$$
$$\leq \sqrt{n} + \sqrt{n} + (K-1)(\sqrt{n} + \sqrt{n}) = 2\sqrt{n}K. \tag{25}$$

$\square$

# I   Implementation of GFSA

The pseudo code of our GFSA is shown in Algorithm 1. For implementation, $w_0$ and $w_1$ can be set as hyperparameters optionally.

---
**Algorithm 1** PyTorch-style pseudocode for GFSA
---
```
w_0 = torch.zeros(h)
w_1 = torch.ones(h)
w_K = torch.zeros(h)
I = torch.eyes(n)[None, None, ...]

def GFSA (att, K)
  att:  original self-attention
  att_K: high order term
  att_K = att + (K-1) * (torch.mm(att,att)-att)
  gf_att:  GFSA attention
  gf_att = w_0[None, :, None, None] * I
          + w_1[None, :, None, None] * att
          + w_K[None, :, None, None] * att_K
return gf_att
```
---

# J   Comparison with Actual and Approximated High-order Terms

To compare the impact of the actual $\bar{\boldsymbol{A}}^K$ and the approximated $\bar{\boldsymbol{A}}^K$ in terms of accuracy, we experimented with BERT on GLUE and the results are summarized in Table 9. $\text{BERT}_{\text{BASE}}+\bar{\boldsymbol{A}}^K$ denotes using the exactly calculated $\bar{\boldsymbol{A}}^K$ instead of the approximated $\bar{\boldsymbol{A}}^K$.

Table 9: Comparison of performance using the exactly calculated $\bar{\boldsymbol{A}}^K$ vs. the approximated $\bar{\boldsymbol{A}}^K$ for GLUE tasks

| Datasets | #Params | CoLA | SST-2 | MRPC | QQP | STS-B | MNLI-m/mm | QNLI | RTE | **Avg** |
|---|---|---|---|---|---|---|---|---|---|---|
| BERT$_{\text{BASE}}$ [16] | 110M | 56.79 | 93.81 | 88.70 | 88.32 | 88.16 | 84.96/84.15 | 91.63 | 66.06 | 82.51 |
| + GFSA (approximated $\bar{\boldsymbol{A}}^K$) | 110M | 59.56 | 94.15 | 90.60 | 88.46 | 88.33 | 85.12/85.06 | 91.95 | 68.95 | 83.58 |
| + GFSA (actual $\bar{\boldsymbol{A}}^K$) | 110M | 59.85 | 94.27 | 89.80 | 88.43 | 88.32 | 84.95/84.89 | 91.76 | 68.23 | 83.39 |

# K    Natural Language Understanding

## K.1    Detailed Experimental Settings

We integrate GFSA into 3 pre-trained large language models: BERT, ALBERT, and RoBERTa. We evaluate them on the GLUE benchmark, which includes 3 categories of natural language understanding tasks: i) single-sentence tasks CoLA and SST-2; ii) similarity and paraphrasing tasks MRPC, QQP, and STS-B; iii) natural language inference tasks MNLI, QNLI, and RTE. For MNLI task, we experiment on both the matched (MNLI-m) and mismatched (MNLI-mm) versions. Following Devlin et al. [16], we report Matthews correlation for CoLA, F1 scores for QQP and MRPC, Spearman correlations for STS-B, and accuracy scores for the other tasks. For each task, we select the best hyperparameters for GFSA, and the other hyperparameters are fixed. We compare our GFSA with ContraNorm [28], one of the related methods that address oversmoothing. We finetune ContraNorm with the recommended hyperparameters in Guo et al. [28]. We initialize with a pre-trained language model and finetune with added GFSA for 5 epochs.

**Dataset.**    The benchmark datasets we used are listed below.

- **CoLA.** The Corpus of Linguistic Acceptability [85] consists of English acceptability judgments drawn from books and journal articles. The target task is a binary classification task, and each sentence is determined to be grammatically acceptable or not.

- **SST-2.** The Stanford Sentiment Treebank [73] is a dataset in which each sentence is sourced from movie reviews and accompanied by human annotations of their sentiment. The target task is to classify binary sentiments for a single sentence.

- **MRPC.** The Microsoft Research Paraphrase Corpus [17] is a corpus of sentence pairs, which are automatically extracted from online news sources and annotated by humans. The target is to determine whether the sentences in the pair are semantically equivalent.

- **QQP.** The Quora Question Pairs [11] dataset is a collection of question pairs from the community question-answering website Quora. The target is to determine whether the questions in the pair are semantically equivalent.

- **STS-B.** The Semantic Textual Similarity Benchmark [8] is a collection of sentence pairs drawn from news headlines, video and image captions, and natural language inference data with human annotation. The target is a regression task to predict a similarity score from 0 to 5.

- **MNLI.** The Multi-Genre Natural Language Inference Corpus [87] is a crowdsourced collection of sentence pairs with textual entailment annotations. Given a premise sentence and a hypothesis sentence, the task is to predict whether the premise entails the hypothesis (entailment), contradicts the hypothesis (contradiction), or neither (neutral). The standard test set consists of private labels from the authors and evaluates both the matched (in-domain) and mismatched (cross-domain) sections.

- **QNLI.** The Stanford Question Answering [82] dataset is a question-answering dataset consisting of question-paragraph pairs, where one of the sentences in the paragraph contains the answer to the corresponding question written by an annotator. The task is to determine whether the context sentence contains the answer to the question.

- **RTE.** The Recognizing Textual Entailment [5] dataset comes from a series of annual textual entailment challenges. The target task is a binary entailment classification task.

**BERT.**    BERT [16] consists with 12 layers, 12 heads, 768 hidden size, 512 maximum sequence length, and MLP dimension of 3072.

**ALBERT.**    ALBERT [40] consists of 12 layers, 12 heads, 768 hidden dimensions, 512 maximum sequence length, 128 embedding dimensions, and MLP dimension of 3072.

**RoBERTa.**    RoBERTa [44] consists of 12 layers, 12 heads, 768 hidden size, 514 maximum sequence length, and MLP dimension of 3072.

**Training.** For implementation, we adopt HuggingFace framework. We trained all models with 5 epochs with 32 batch size. The linear learning rate decay is used and initial learning rate is set to $2 \times 10^{-5}$. We use AdamW [46] optimizer, and weight decay is set to 0. All models are trained on 1 GPU and of NVIDIA RTX A5000 24GB.

### K.2 Sensitivity to $K$

In this section, we explore the influence of the polynomial order, denoted as $K$, in our GFSA, conducting experiments on BERT$_{\text{BASE}}$ finetuned with GLUE tasks. We search for values of $K$ from 2 to 10, and the results are presented in Table 10. For each dataset, there is an optimal $K$ and the performance of models using GFSA is generally robust to changes in $K$.

Table 10: Sensitivity results on various $K$ with BERT$_{\text{BASE}}$ finetuned on GLUE tasks

| $K$ | CoLA | SST2 | MRPC | QQP | STSB | MNLI-m | MNLI-mm | QNLI | RTE |
|---|---|---|---|---|---|---|---|---|---|
| 2 | 57.83 | **94.15** | **90.60** | 88.41 | 88.27 | 84.96 | 84.90 | 91.74 | 68.23 |
| 3 | 58.56 | 93.46 | 89.77 | 88.41 | **88.33** | 85.08 | 84.75 | 91.78 | 68.59 |
| 4 | **59.56** | 93.46 | 89.77 | 88.45 | 88.29 | 85.06 | **85.06** | 91.76 | 68.59 |
| 5 | 58.10 | 93.58 | 90.07 | 88.39 | 88.29 | 84.87 | 84.99 | 91.85 | 68.95 |
| 6 | 59.40 | 93.58 | 90.40 | 88.29 | 88.27 | 84.93 | 84.97 | 91.69 | 68.23 |
| 7 | 59.12 | 94.04 | 90.48 | 88.43 | 88.26 | **85.12** | 84.94 | 91.82 | 68.94 |
| 8 | 58.58 | 93.69 | 90.12 | **88.46** | 88.24 | 84.92 | 84.81 | **91.95** | **68.95** |
| 9 | 58.88 | 93.46 | 89.54 | 88.41 | 88.26 | 85.06 | 91.67 | 67.87 | 85.04 |
| 10 | 59.31 | 93.35 | 89.98 | 88.41 | 88.30 | 84.84 | 91.73 | 68.59 | 84.93 |

## L  Causal Language Modeling

### L.1  Detailed Experimental Settings

**Dataset.**  The benchmark datasets we used are listed below.

- **PTB.** Penn Treebank [47] dataset is a collection of text documents that have been extensively annotated with linguistic information, primarily syntactic and grammatical structures.
- **WikiText.** WikiText [50] dataset is a collection of over 100 million tokens extracted from the set of verified good and featured articles on Wikipedia. Compared to the preprocessed version of Penn Treebank (PTB), WikiText-2 is over 2 times larger and WikiText-103 is over 110 times larger.

**GPT2.**  GPT2 [61] is a Transformer pretrained on a very large corpus of English data in a self-supervised fashion without any human labelling on dataset. It automatically generate inputs and labels from those texts, and trained to guess the next word in sentences. For implementation, we adopt HuggingFace Framework [4]. For all experiments, GPT2 has 12 layers with 12 attention heads, 768 hidden size and 1024 maximum sequence length, resulting in a total of 117 million parameters.

**Training.**  We finetune GPT2 with 4 batch size, $5 \times 10^{-5}$ learning rate and linear learning weight decay using adamW [46] optimizer. We also apply dropout with probability 0.1. Following [93], we train models for 15 epochs with PTB, 4 epochs with WikiText-103 and 10 epochs with WikiText-2. We use sensitivity metric, i.e., perplexity, which is a commonly used metric to evaluate the performance of language models, particularly in language modeling and text generation tasks. perplexity measures how well a language modeling can predict a sequence of words in a given text or a test dataset. All the experiments are conducted on 1 GPU and of NVIDIA RTX 3090 24GB.

### L.2  Sensitivity to $K$

We conducted a sensitivity study on $K$ of GPT-2 across all datasets, and the results are presented in Table 11. For PTB and WikiText-2, GFSA exhibits the best performance when $K$ is high, typically

---

[4] https://github.com/huggingface/transformers

around 8 or 9. However, for WikiText-103, GFSA achieves the best perplexity when $K$ is small, specifically when $K$ is 3 or 4.

Table 11: Results comparison on GPT-2 finetuned with GFSA

| Method | #Params | PTB | WikiText-2 | WikiText-103 |
|---|---|---|---|---|
| GPT2 [61] | 117M | 19.513 | 20.966 | 15.939 |
| GPT2 + GFSA($K = 2$) | 117M | 19.459 | 20.929 | 15.920 |
| GPT2 + GFSA($K = 3$) | 117M | 19.456 | 20.927 | **15.919** |
| GPT2 + GFSA($K = 4$) | 117M | 19.453 | 20.927 | **15.919** |
| GPT2 + GFSA($K = 5$) | 117M | 19.452 | 20.925 | 15.920 |
| GPT2 + GFSA($K = 6$) | 117M | 19.451 | 20.925 | 15.920 |
| GPT2 + GFSA($K = 7$) | 117M | **19.450** | 20.925 | 15.921 |
| GPT2 + GFSA($K = 8$) | 117M | **19.450** | 20.924 | 15.921 |
| GPT2 + GFSA($K = 9$) | 117M | **19.450** | **20.923** | 15.921 |

# M   Image Classification

## M.1   Detailed Experimental Settings

Our code is implemented based on the timm library [86]. In the case of our training recipe, it is the same as experimental setting of Wang et al. [80] that follows the training recipes of Touvron et al. [74] and Touvron et al. [75]. To apply our GFSA to existing base models such as DeiT, Cait, and Swin, we consider a range of $K$ between 2 and 5. For 12-layer DeiT, we follow the same hyperparameters from Wang et al. [80]. We set the dropout rate to 0 and 0.2 for 12-layer and 24-layer DeiT, respectively. For CaiT, we apply our GFSA on only to the patch embedding layer. All other hyper-parameters are kept consistent with the original papers of DeiT [74], CaiT [75] and, Swin [45]. All models are trained on NVIDIA RTX 3090 24GB.

## M.2   FLOPs & Throughput

In Table 12, we report the number of FLOPs and throughput. With GFSA plugged in, the FLOP count is either the same or no different. For DeiT-S with 24 layers, which shows a slight FLOP increase with GFSA plugged in. However, for the rest of the settings, the models have the same number of Flops. For throughput, it tends to decrease because calculating the high-order term is an additional cost.

Table 12: Experimental evalutation of GFSA plugged into DeiT-S, CaiT-S, and Swin-S

| Backbone | Method | Input Size | #Layers | #Params | #FLOPs | #Throughput | Top-1 Acc |
|---|---|---|---|---|---|---|---|
| DeiT | DeiT-S | 224 | 12 | 22.0M | 4.57G | 856.07 | 79.8 |
| | DeiT-S + GFSA | 224 | 12 | 22.0M | 4.57G | 614.54 | **81.1** ($\uparrow$ 1.3) |
| | DeiT-S | 224 | 24 | 43.3M | 9.09G | 423.68 | 80.5 |
| | DeiT-S + GFSA | 224 | 24 | 43.3M | 9.10G | 314.75 | **81.5** ($\uparrow$ 1.0) |
| CaiT | CaiT-S | 224 | 24 | 46.9M | 9.34G | 574.66 | 82.6 |
| | CaiT-S + GFSA | 224 | 24 | 47.0M | 9.34G | 406.96 | **82.8** ($\uparrow$ 0.2) |
| Swin | Swin-S | 224 | 24 | 49.6M | 8.75G | 912.38 | 82.9 |
| | Swin-S + GFSA | 224 | 24 | 49.6M | 8.75G | 714.60 | **83.0** ($\uparrow$ 0.1) |

## M.3   Sensitivity to $K$

We also perform the sensitivity analysis for $K$. Tables 13 and 14 show the results of sensitivity analysis for DeiT-S and CaiT-S with GFSA plugged in. For 12-layer DeiT-S, GFSA performance of 81.12 is highest when $K = 3$. When the GFSA has a $K$ of 2, the performance is worse than the

original DeiT-S, but when the $K$ is 3 or higher, the performance is better than the original DeiT-S, and most surprisingly, the performance is better than the 24-layer DeiT-S.

CaiT-S shows the highest performance of 82.84 when $K = 4$. For CaiT-S, the accuracy is slightly lower than that of the original CaiT-S when $K = 2$, but it starts to exceed the accuracy of CaiT-S when $K$ is 3 or higher.

Table 13: Sensitivity to $K$ for 12-layer DeiT-S + GFSA

| $K$ | 2 | 3 | 4 | 5 |
|---|---|---|---|---|
| Top-1 Acc (%) | 79.27 | **81.12** | 80.86 | 81.07 |

Table 14: Sensitivity to $K$ for 24-layer CaiT-S + GFSA

| $K$ | 2 | 3 | 4 |
|---|---|---|---|
| Top-1 Acc (%) | 82.54 | 82.65 | **82.84** |

## M.4  Full Experimental Results

In Table 15, we consider all three classes, CNN only, CNN + Transformer, and pure Transformer, to compare more different models than in Table 3. In particular, in the Transformer category, we only test with lightweight models with similar number of parameters, such as ViT-S and DeiT-S. Compared to existing techniques, the improvements by GFSA already surpasses LayerScale (0.7%) [75], LateInsertion (0.6%) [75], and HAT [4] (1.38%).

Table 15: Compared with state-of-the-art models on ImageNet-1k dataset. The number in (↑) indicates the performance improvement over the base model.

| Category | Method | Input Size | #Layers | #Params | Top-1 Acc |
|---|---|---|---|---|---|
| CNN | ResNet-152 [29] | 224 | 152 | 230M | 78.1 |
| | DenseNet-201 [34] | 224 | 201 | 77M | 77.6 |
| CNN + Transformer | CVT-21 [88] | 224 | 21 | 32M | 82.5 |
| | Refiner [102] | 224 | 16 | 86M | 81.2 |
| Transformer | ViT-S/16 [19] | 224 | 12 | 49M | 78.1 |
| | ViT-B/16 [19] | 224 | 12 | 86M | 79.8 |
| | DeiT-S [74] | 224 | 12 | 22M | 79.8 |
| | DeiT-S + LayerScale [75] | 224 | 12 | 22M | 80.5 |
| | DeiT-S + LateInsertion [75] | 224 | 12 | 22M | 80.5 |
| | DeiT-S + ClassAttention [75] | 224 | 12 | 22M | 80.6 |
| | DeiT-S + AttnScale [80] | 224 | 12 | 22M | 80.7 |
| | DeiT-S + FeatScale [80] | 224 | 12 | 22M | 80.9 |
| | DeiT-S + HAT [4] | 224 | 12 | 22M | 80.9 |
| | DeiT-S + Diverse [10] | 224 | 12 | 22M | 80.6 |
| | DeiT-S + ContraNorm [28] | 224 | 12 | 22M | 80.4 |
| | Swin-S [45] | 224 | 12 | 50M | 82.9 |
| | T2T-ViT-24 [96] | 224 | 24 | 64M | 82.3 |
| | DeepViT-24B [101] | 224 | 24 | 36M | 80.1 |
| | DeiT-S [74] | 224 | 24 | 43M | 80.5 |
| | CaiT-S [75] | 224 | 24 | 47M | 82.6 |
| | DeiT-S + DiversePatch [25] | 224 | 24 | 44M | 82.2 |
| | DeiT-S + LayerScale [75] | 224 | 24 | 44M | 82.4 |
| | DeiT-S + AttnScale [80] | 224 | 24 | 44M | 81.1 |
| | DeiT-S + FeatScale [80] | 224 | 24 | 44M | 81.3 |
| | DeiT-S + ContraNorm [28] | 224 | 24 | 43M | 80.7 |
| GFSA | DeiT-S + GFSA | 224 | 12 | 22M | **81.1** (↑ 1.3) |
| | DeiT-S + GFSA | 224 | 24 | 43M | **81.5** (↑ 1.0) |
| | CaiT-S + GFSA | 224 | 24 | 47M | **82.8** (↑ 0.2) |
| | Swin-S + GFSA | 224 | 12 | 50M | **83.0** (↑ 0.1) |

## M.5 Additional Comparison with SOTA Models

Our main experiment aims to determine whether introducing the GFSA layer would help improve performance in a base model, such as DeiT. We also compare our method with the recent models: SpectFormer [58], SVT [57], NeuTRENO [52], FNet [42], and GFNet [64]. We use a 12-layer setup to ensure a fair comparison in Table 16.

GFNet [64] can reduce the number of parameters, but there is a performance penalty. However, the performance improvement of DeiT-S+GFSA is relatively greater than DeiT-S compared to other models. SpectFormer [58] and SVT [57] have advantages in calculation amount and model complexity, and performance is improved over DeiT-S, but Top-1 and Top-5 accuracies are lower than those using GFSA. Additionally, NeuTRENO [52] also improves as much as GFSA compared to DeiT-S, but GFSA still has higher Top-1 accuracy.

Table 16: Compared with state-of-the-art models on ImageNet-1k

| Method | Input Size | #Layers | #Params | Top-1 Acc | Top-5 Acc |
|---|---|---|---|---|---|
| DeiT-S | 224 | 12 | 22M | 79.8 | 95.0 |
| Fnet-XS [42] | 224 | 12 | 20M | 71.2 | - |
| GFNet-XS [64] | 224 | 12 | 16M | 78.6 | 94.2 |
| SpectFormer-XS [58] | 224 | 12 | 20M | 80.2 | 94.7 |
| SVT-XS [57] | 224 | 12 | 20M | 79.9 | 94.5 |
| DeiT-S + NeuTRENO [52] | 224 | 12 | 20M | 80.7 | 95.4 |
| DeiT-S + GFSA | 224 | 12 | 22M | **81.1** | 95.4 |

## M.6 Additional Experiments with Guo et al. [28]'s setting

To make a fair comparison with ContraNorm [28], one of the related studies that mitigates over-smoothing, we run additional experiments to match their experimental setup.

**Setting.** We follow the training recipe used by Guo et al. [28], which is a slightly modified version of Touvron et al. [74]'s recipe. Guo et al. [28] use AdamW optimizer with cosine learning rate decay. We select the DeiT-T and DeiT-S for ImageNet-1k. "T" and "S" denote tiny and small model sizes, respectively. For all experiments, the image size is set to be 224x224. We train each model for 300 epochs and the batch size is set to 1024. For ContraNorm, we train with their recommended hyperparameters. All models are trained on 4 GPUs and of NVIDIA RTX A6000 48GB.

**Results.** In Table 17, DeiT-T and DeiT-S with GFSA outperform vanilla DeiT-T and DeiT-S in all layer settings. GFSA improves the performance of DeiT-T with 12 layers by 1.52%. The largest gain is a 4.88% improvement on 16-layer DeiT-T. This shows that the effect of GFSA is larger than the effect of ContraNorm. For DeiT-S with 16 layers, surprisingly, GFSA is able to increase the performance by 80.83%, meaning that GFSA brings benefits with a 3.23% improvement.

Table 17: Experiment results on ImageNet-1k

| Method | #Layers=12 | #Layers=16 | #Layers=24 |
|---|---|---|---|
| DeiT-T | 76.52 | 75.34 | 76.76 |
| DeiT-T + ContraNorm | 77.03 | 78.72 | 78.12 |
| DeiT-T + GFSA | **77.68** | **79.02** | **78.64** |
| DeiT-S | 77.32 | 78.25 | 77.69 |
| DeiT-S + ContraNorm | 77.80 | 79.04 | 78.67 |
| DeiT-S + GFSA | **79.86** | **80.83** | **79.15** |

**Sensitivity to $K$.** In Table 18, we experiment with a sensitivity analysis for $K$. For DeiT-T, the performance of GFSA generally improves when $K$ is 4 or 5. On the other hand, GFSA performs better at lower $K$ for settings that are layers 16 and 24 for DeiT-S.

Table 18: Varying $K$ for DeiT-T and DeiT-S

| Method | $K$ | #Layers=12 | #Layers=16 | #Layers=24 |
|--------|-----|------------|------------|------------|
| DeiT-T + GFSA | 2 | 76.92 | 78.14 | 78.40 |
| DeiT-T + GFSA | 3 | 77.41 | 77.76 | 78.09 |
| DeiT-T + GFSA | 4 | 77.01 | **79.02** | **78.64** |
| DeiT-T + GFSA | 5 | **77.68** | 78.14 | **78.64** |
| DeiT-S + GFSA | 2 | 79.84 | **80.83** | **79.15** |
| DeiT-S + GFSA | 3 | 79.85 | 79.39 | 79.07 |
| DeiT-S + GFSA | 4 | **79.86** | 79.44 | 79.10 |

# N  Automatic Speech Recognition

## N.1  Detailed Experimental Settings

**Dataset.**  We conduct experiments on the LibriSpeech [5] dataset [55], which consists of audio recordings paired with their transcriptions. The LibriSpeech dataset has approximately 1,000 hours of read English speech with a sampling rate of 16 kHz. We keep the original 16,000Hz sampling rate and compute 80-dim log-Mel filterbanks for a 25ms sliding window, strided by 10ms. The filterbank features are then normalized to zero mean and unit variance per input sequence. For implementation, we follow the recipes of SpeechBrain [65].

**Evaluation Metric.**  Word error rate (WER (%)) is derived from the Levenshtein distance and compares a reference to a hypothesized word-level transcription. It is calculated by summing the number of word insertions, deletions, substitutions and dividing it by the total number of words in the reference transcription.

**Vanilla Transformer.**  We use a vanilla Transformer to apply our GFSA. For implementation, we use a SpeechBrain [65] framework. The vanilla Transformer consists of i) 1D convolution to perform striding, ii) Transformer encoder with 12 layers, 4 heads, embedding dimension of 512, MLP dimension of 2048, and post-LayerNorm iii) decoder with 6 layers, 4 heads, embedding dimension of 512, MLP dimension of 2048, joint beamsearch, and iv) external Transformer language model with 12 layers, 12 heads, embedding dimension of 768, and MLP dimension of 3072.

**Branchformer.**  We use one of the SOTA models, Branchformer [59] to plug-in our GFSA. Branchformer has two parallel branches, one for capturing global interactions using attention and the other for more localized context using convolutional gating MLP. The Branchformer architecture for speech recognition consists of i) 1D convolution to perform striding, ii) Branchformer encoder with 18 layers, 8 heads, embedding dimension of 512, and MLP dimension of 3072, iii) decoder with 6 layers, 8 heads, embedding dimension of 512, a convolutional spatial gating unit (CSGU) dimension of 3072, joint beamsearch, and iv) external Transformer language model with 12 layers, 12 heads, embedding dimension of 768, and MLP dimension of 3072.

**Training.**  We follow a training recipe from SpeechBrain [65]. The standard LibriSpeech validation sets (dev-clean and dev-other) are used to tune all parameters and select the best models. Test sets (test-clean and test-other) are used only to report final WER performance. We train the pure Transformer for 100 epochs and the Branchformer for 120 epochs with a batch size of 16. We use a data augmentation method on all models using SpecAugment [56]. SpecAugment applies time and frequency masking as well as time warping to the input spectrum. For Branchformer, we use AdamW [46] optimizer with 0.9 and 0.98 coefficients for computing running averages of gradient and its square. The learning rate and weight decay in all models are 0.0008 and 0.01, respectively. We use a connectionist temporal classification (CTC) loss [26, 100]. We also apply dropout with probability 0.1 and label smoothing with weight 0.1 to mitigate overfitting. We fix the random seed as 74443 on all experiments. All models are trained on 1 GPU and of NVIDIA RTX A6000 48GB.

---

[5]http://www.openslr.org/12

**Hyperparameters.** In Table 19, we describe main hyperparameters used in the automatic speech recognition task. For Transformer+GFSA and Branchformer+GFSA, we also report the best $K$ hyperparameter.

Table 19: Main hyperparameters used in ASR

| Model | Experimental Setting |
|---|---|
| Transformer | Encoder: Transformer (12 layers)
Decoder: Transformer (6 layers) + (CTC/ATT joint) beamsearch + TransformerLM
Augmentation: SpecAugment
Features: 40 fbanks
Pretraining: no
Dropout: 0.1
Batchnorm: yes
Number of epochs: 100
Batch size: 32
Learning rate: 0.0008
LR scheduler: Noam
Optimizer: Adam
Loss: CTC + KLdiv (Label Smoothing loss)
CTC weight: 0.3 |
| Transformer+GFSA | Encoder: Transformer (12 layers)
Decoder: Transformer (6 layers) + (CTC/ATT joint) beamsearch + TransformerLM
Augmentation: SpecAugment
Features: 40 fbanks
Pretraining: no
Dropout: 0.1
Batchnorm: yes
Number of epochs: 100
Batch size: 32
Learning rate: 0.0008
LR scheduler: Noam
Optimizer: Adam
Loss: CTC + KLdiv (Label Smoothing loss)
CTC weight: 0.3
$K$: 2 |
| Branchformer | Encoder: Branchformer
Decoder: Transformer (6 layers) + (CTC/ATT joint) beamsearch + TransformerLM
Augmentation: SpecAugment
Features: 40 fbanks
Pretraining: no
Dropout: 0.1
Batchnorm: yes
Number of epochs: 120
Batch size: 16
Learning rate: 0.0008
LR scheduler: Noam
Optimizer: AdamW with coefficients 0.9 and 0.98
Loss: CTC + KLdiv (Label Smoothing loss)
CTC weight: 0.3 |
| Branchformer+GFSA | Encoder: Branchformer
Decoder: Transformer (6 layers) + (CTC/ATT joint) beamsearch + TransformerLM
Augmentation: SpecAugment
Features: 40 fbanks
Pretraining: no
Dropout: 0.1
Batchnorm: yes
Number of epochs: 120
Batch size: 16
Learning rate: 0.0008
LR scheduler: Noam
Optimizer: AdamW with coefficients 0.9 and 0.98
Loss: CTC + KLdiv (Label Smoothing loss)
CTC weight: 0.3
$K$: 3 |

## N.2 Training Curve

We compare the training and validation curves for LibriSpeech 100h in Fig. 8. The training loss curve of GFSA is lower than the pure Transformer. GFSA stabilizes the loss curve of pure Transformer slightly earlier.

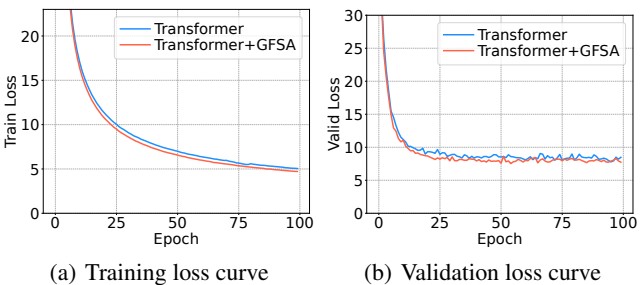

(a) Training loss curve       (b) Validation loss curve

Figure 8: Training curve on LibriSpeech 100h

# O  Graph-level Tasks

## O.1  Detailed Experimental Settings

**Experimental settings for Graphormer.**  We describe benchmark datasets and Graphormer as the backbone model we used. ZINC [35] is the most popular real-world molecular dataset to predict graph property regression for constrained solubility, an important chemical property for designing generative GNNs for molecules. Uniform sampling is adopted for data splitting. We use a ZINC-subset of small-scale dataset. PCQM4M-LSC  [33] is 2D molecular graphs, which is one of the most practically relevant quantum chemical properties of molecule science. The task is to predict density functional theory (DFT)-calculated HOMO-LUMO energy gap of molecules given their graphs. PCQM4M-LSC is unprecedentedly large in scale comparing to other labeled graph-level prediction datasets, which contain more than 3.8M graphs. We use PCQM4M and PCQM4Mv2 large-scale datasets.

Following Graphormer [94], we use Graphormer for PCQM4M and Graphormer$_{\text{SLIM}}$ for ZINC. Graphormer consists of 12 encoder layers, 80 encoder embedding dimension, and 768 MLP dimension. It employs 32 encoder heads and 24 hidden dimension for each head. Graphormer$_{\text{SLIM}}$ consists of 12 encoder layers, 80 encoder embedding dimension, and 80 MLP dimension. It employs 8 encoder heads and 10 hidden dimension for each head. We use adamW [46] optimizer with 0.9 and 0.999 coefficients for running averages of gradient and its square, and use Mean Absolute Error (MAE) as loss function. We use polynomial learning rate decay, with initial learning rate set to $2 \times 10^{-4}$ and end learning rate set to $1 \times 10^{-9}$. For ZINC, we set batch size as 256, max epochs as 10k, and warm-up stage step as 40k. For PCQM4M and PCQM4Mv2, we set batch size as 1024, max epochs as 300, and warm-up stage step as 60k. All models are trained on 1 GPU and of NVIDIA RTX 3090 24GB. We conduct experiments with 4 different seeds.

**Experimental settings for GPS and Graph-ViT.**  We use various benchmark datasets to experiment with our GFSA on GPS [63] and Graph-ViT [30]: Peptide-func and Peptide-struct from Long-Range Graph Benchmark (LRGB) [21], MNIST, CIFAR10, and ZINC from Benchmarking GNNs [22], and Moltox21 and Molhiv from OGB [32]. We fix all hyperparameters of GPS and Graph-ViT as recommended in their paper to ensure a fair comparison. To plug GFSA into GPS, we replace the self-attention module of GPS with GFSA. For Graph-ViT, we apply GFSA instead of the Hadamard self-attention mechanism and compare it with this self-attention method. We conduct experiments on GPS and Graph-ViT in their open code frameworks for a fair comparison:

- GPS: `https://github.com/rampasek/GraphGPS`

- Graph-ViT: `https://github.com/XiaoxinHe/Graph-ViT-MLPMixer`

### O.2 Experimental Results with Standard Deviation

We conduct experiments following the experimental environments of Graphormer [94] using 4 different seeds. Due to space constraints, only the mean values are reported in Tables 4 and 5. In Tables 20 and 21, we report the results with mean and standard deviations.

Table 20: Experimental results and number of parameters on ZINC

| Method | #Params | MAE |
|---|---|---|
| Graphormer | 500K | $0.1240_{\pm 0.006}$ |
| Graphormer + GFSA | 500K | $\mathbf{0.1189}_{\pm 0.002}$ |

Table 21: Experimental results and number of parameters on PCQM4M and PCQM4Mv2

| Method | #Params | PCQM4M | | PCQM4Mv2 | |
|---|---|---|---|---|---|
| | | Train | Validate | Train | Validate |
| Graphormer | 48.3M | $0.0535_{\pm 0.038}$ | $0.1286_{\pm 0.016}$ | $0.0250_{\pm 0.000}$ | $0.0862_{\pm 0.000}$ |
| Graphormer + GFSA | 48.3M | $\mathbf{0.0312}_{\pm 0.001}$ | $\mathbf{0.1193}_{\pm 0.000}$ | $\mathbf{0.0249}_{\pm 0.000}$ | $\mathbf{0.0860}_{\pm 0.000}$ |

## P  Code Defect Detection

### P.1  Detailed Experimental Settings

**Dataset.**   We use Devign dataset provided by [105], which is a binary classification task to evaluate whether a C language code is vulnerable to software systems or not.

**Implementation.**   We build our experiments on top of the open-sourced code [6] and recipes provided by Wang et al. [84].

**RoBERTa.**   RoBERTa [44] is an encoder-only model trained with masked language modeling on code. All hyperparameters are consistent with the training method in the source code of Wang et al. [84].

**PLBART.**   PLBART [2] is an encoder-decoder model based on BART [43] architecture. PLBART can support understanding and generation tasks. All hyperparameters are consistent with the training method in the source code of Wang et al. [84].

**CodeBert.**   CodeBERT [23] is a model trained on masked language modeling and replaced token detection. CodeBERT is a bimodal pretrained model based on Transformer with 12 layers for programming language and natural language. All hyperparameters are consistent with the training method in the source code of Wang et al. [84].

**CodeT5.**   CodeT5 is an encoder-decoder framework with the same architecture as T5 [62]. It aims to derive generic representations for programming language and natural language via pre-training on unlabeled source code. CodeT5-small has 6 encoder layers, 6 decoder layers, 8 attention heads, 512 dimensional hidden states, and 60M parameters. The other models have 12 encoder layers, 12 decoder layers, 12 attention heads, 768 dimensional hidden states, and 220M parameters. All hyperparameters are consistent with the training method in the source code of Wang et al. [84].

**Training.**   The pre-trained models mentioned above are applied to this downstream task. We add GFSA directly on top of self-attention. We finetune baselines and GFSA models for 10 epochs with a batch size of 16. We use early stopping strategy with a patience of 2. Models generate binary labels from unigram sequences at the decoder for defect detection task. We employ accuracy for evaluating the code defect detection task. All models are trained on 1 GPU and of NVIDIA RTX A6000 48GB.

---

[6] https://github.com/salesforce/CodeT5

### P.2 Case Study

In Listing 1, we show one of case for code snippets of defects in QEMU [7] that CodeT5-base does not predict correctly, but that *only* CodeT5-base+GFSA predicts. The commit message [8] for this case is as follow:

> Needed for changing `cpu_has_work()` argument type to CPUState, used in `h_cede()`.

`h_cede()` is the hypercall that asks the hypervisor to shut down the CPU. Previously, this hypercall simply passed the CPUID, so the hypervisor did not know what state the CPU was in. This change allows the hypervisor to know whether the CPU is actually performing work. If the CPU is performing a task, the hypervisor waits for the CPU to complete the task.

In this context, accurately predicting defects like the one above is very important, and applying GFSA to CodeT5-base helps in terms of performance improvement.

```
1 @@ -204,7 +204,7 @@ static target_ulong put_tce_emu(sPAPRTCETable *
    tcet, target_ulong ioba,
2 - static target_ulong h_put_tce(CPUPPCState *env, sPAPREnvironment *
    spapr
3 + static target_ulong h_put_tce(PowerPCCPU *cpu, sPAPREnvironment *
    spapr
4                                   , target_ulong opcode, target_ulong *
    args)
5  {
6    target_ulong liobn = args[0];
7    target_ulong ioba = args[1];
8    target_ulong tce = args[2];
9    VIOsPAPRDevice *dev = spapr_vio_find_by_reg(spapr->vio_bus, liobn);
10   VIOsPAPR_RTCE *rtce;
11   if (!dev) {
12     hcall_dprintf("LIOBN 0x" TARGET_FMT_lx " does not exist\n", liobn)
     ;
13     return H_PARAMETER;
14   }
15   ioba &= ~(SPAPR_VIO_TCE_PAGE_SIZE - 1);
16   #ifdef DEBUG_TCE
17     fprintf(stderr, "spapr_vio_put_tce on %s ioba 0x" TARGET_FMT_lx "
     TCE 0x" TARGET_FMT_lx "\n", dev->qdev.id, ioba, tce);
18   #endif
19   if (ioba >= dev->rtce_window_size) {
20     hcall_dprintf("Out-of-bounds IOBA 0x" TARGET_FMT_lx "\n", ioba);
21     return H_PARAMETER;
22   }
23   rtce = dev->rtce_table + (ioba >> SPAPR_VIO_TCE_PAGE_SHIFT);
24   rtce->tce = tce;
25   return H_SUCCESS;
26 }
```

Listing 1: An example commit history for defects in Devign dataset

## Q    Code Clone Detection

### Q.1    Detailed Experimental Settings

**Dataset.** Code clone detection aims to measure the similarity between two code snippets and predict whether they have the same functionality. We experiment with the Java data provided by Wang et al. [83].

---

[7] https://www.qemu.org
[8] https://github.com/qemu/qemu/commit/b13ce26d3e8c6682044ae84920f2417b30ce356b

**Implementation.**    We build our experiments on top of the open-sourced code [9] and recipes provided by Wang et al. [84].

**Training.**    We finetune both CodeT5 and CodeT5+GFSA for one epoch with a batch size of 16. We also use early stopping with patience of 2. CodeT5 and CodeT5+GFSA encode source code and take the representation to calculate similarity of two code snippets. We employ F1 score for evaluating this task. All models are trained on 1 GPU and of NVIDIA RTX A6000 48GB.

### Q.2    Experiment Result

Table 22 shows results comparing CodeT5 and CodeT5 with GFSA. The result shows that by using our GFSA, CodeT5 models improve their performance. CodeT5-small+GFSA provides a 0.61% improvment over Code5T-small.

Table 22: Results on the code clone detection task

| Method | Clone F1 |
|---|---|
| CodeT5-small [84] | 94.36 |
| CodeT5-small + GFSA | **94.94** (↑ 0.61%) |
| CodeT5-base [84] | 94.31 |
| CodeT5-base + GFSA | **94.92** (↑ 0.64%) |

## R    Time Complexity and Empirical Runtime Analysis

**Time Complexity.**    The time complexity of original self-attention is $\mathcal{O}(n^2 d)$. But our GFSA has a high order term. Therefore, the time complexity of GFSA has $\mathcal{O}(n^2 d + n^3)$. If $n$ is smaller than $d$, the time complexity approaches $\mathcal{O}(n^2 d)$, which is the complexity of original self-attention.

**Empirical Runtime Analysis.**    We report the training time of various methods with GFSA in Tables 23 to 28. In general, the training time of methods with GFSA is slightly longer than that of existing methods. For example, the Transformer for the automatic speech recognition task increases from 190.5 seconds to 191.6 seconds on Librispeech 100h dataset, as increases of only 1 second. Instead of computing higher-order polynomial terms, our GFSA approximates them, with only a small increase in runtime, which is not very significant.

Table 23: Training time (seconds per epoch) on GLUE tasks. $s$ denotes the abbreviation for second. **Avg** denotes the average training time across all tasks.

| Datasets | #Params | CoLA | SST-2 | MRPC | QQP | STS-B | MNLI-m/mm | QNLI | RTE | **Avg** |
|---|---|---|---|---|---|---|---|---|---|---|
| BERT$_{BASE}$ [16] | 110M | 17$s$ | 182$s$ | 17$s$ | 1483$s$ | 24$s$ | 2004$s$ | 580$s$ | 18$s$ | 541$s$ |
| BERT$_{BASE}$ + GFSA | 110M | 19$s$ | 192$s$ | 19$s$ | 1571$s$ | 25$s$ | 2147$s$ | 621$s$ | 20$s$ | 577$s$ |
| ALBERT$_{BASE}$ [40] | 11M | 15$s$ | 188$s$ | 20$s$ | 1506$s$ | 25$s$ | 2072$s$ | 612$s$ | 19$s$ | 557$s$ |
| ALBERT$_{BASE}$ + GFSA | 11M | 16$s$ | 197$s$ | 21$s$ | 1604$s$ | 26$s$ | 2219$s$ | 659$s$ | 21$s$ | 595$s$ |
| RoBERTa$_{BASE}$ [44] | 125M | 17$s$ | 190$s$ | 18$s$ | 1492$s$ | 25$s$ | 2012$s$ | 593$s$ | 18$s$ | 546$s$ |
| RoBERTa$_{BASE}$ + GFSA | 125M | 19$s$ | 200$s$ | 19$s$ | 1580$s$ | 26$s$ | 2151$s$ | 634$s$ | 20$s$ | 581$s$ |

Table 24: Training time (seconds per epoch) on causal language modeling tasks.

| Method | #Params | PTB | WikiText-2 | WikiText-103 | **Avg** |
|---|---|---|---|---|---|
| GPT2 [61] | 117M | 89.1$s$ | 195.7$s$ | 9638.4$s$ | 3307.8$s$ |
| GPT2 + GFSA | 117M | 160.3$s$ | 354.2$s$ | 17424.6$s$ | 5979.7$s$ |

---

[9]https://github.com/salesforce/CodeT5

Table 25: Training time (seconds per epoch) on ImageNet-1k

| Backbone | Method | #Layers | #Params | #FLOPs | #Throughput | Runtime |
|---|---|---|---|---|---|---|
| DeiT | DeiT-S | 12 | 22.0M | 4.57G | 856.07 | 551$s$ |
| | DeiT-S + GFSA | 12 | 22.0M | 4.57G | 614.54 | 814$s$ |
| | DeiT-S | 24 | 43.3M | 9.09G | 423.68 | 1508$s$ |
| | DeiT-S + GFSA | 24 | 43.3M | 9.10G | 314.75 | 1798$s$ |
| CaiT | CaiT-S | 24 | 46.9M | 9.34G | 574.66 | 1530$s$ |
| | CaiT-S + GFSA | 24 | 47.0M | 9.34G | 406.96 | 1624$s$ |
| Swin | Swin-S | 24 | 49.6M | 8.75G | 912.38 | 1897$s$ |
| | Swin-S + GFSA | 24 | 49.6M | 8.75G | 714.60 | 1970$s$ |

Table 26: Training time (seconds per epoch) on graph-level tasks

| Method | ZINC | PCQM4M | PCQM4Mv2 |
|---|---|---|---|
| Graphormer [94] | 9$s$ | 740$s$ | 817$s$ |
| Graphormer + GFSA | 9$s$ | 896$s$ | 955$s$ |

Table 27: Training time (seconds per epoch) on LibriSpeech datasets

| Method | LibriSpeech 100h | LibriSpeech 960h |
|---|---|---|
| Transformer | 190.5$s$ | 3049.3$s$ |
| Transformer + GFSA | 191.6$s$ | 3398.4$s$ |
| Branchformer [59] | 248.5$s$ | 4990.1$s$ |
| Branchformer + GFSA | 254.3$s$ | 4999.3$s$ |

Table 28: Training time (seconds per epoch) on the code defect prediction task

| Method | Runtime |
|---|---|
| RoBERTa [44] | 543.96$s$ |
| RoBERTa + GFSA | 537.79$s$ |
| CodeBERT [23] | 555.28$s$ |
| CodeBERT + GFSA | 561.43$s$ |
| PLBART [2] | 467.80$s$ |
| PLBART + GFSA | 470.19$s$ |
| CodeT5-small [84] | 301.11$s$ |
| CodeT5-small + GFSA | 309.04$s$ |
| CodeT5-base [84] | 362.28$s$ |
| CodeT5-base + GFSA | 373.22$s$ |

# S  Inference Time Analysis

We report the inference time of various methods with GFSA in Tables 29 to 34.

Table 29: Inference time on GLUE tasks. $s$ denotes the abbreviation for second. **Avg** denotes the average training time across all tasks.

| Datasets | #Params | CoLA | SST-2 | MRPC | QQP | STS-B | MNLI-m/mm | QNLI | RTE | **Avg** |
|---|---|---|---|---|---|---|---|---|---|---|
| BERT$_{\text{BASE}}$ [16] | 110M | 1.0$s$ | 1.4$s$ | 1.2$s$ | 48.7$s$ | 1.9$s$ | 15.5$s$ | 10.1$s$ | 1.2$s$ | 10.0$s$ |
| BERT$_{\text{BASE}}$ + GFSA | 110M | 1.1$s$ | 1.4$s$ | 1.2$s$ | 52.3$s$ | 2.0$s$ | 16.8$s$ | 11.0$s$ | 1.3$s$ | 11.0$s$ |
| ALBERT$_{\text{BASE}}$ [40] | 11M | 1.1$s$ | 1.6$s$ | 1.4$s$ | 58.4$s$ | 2.2$s$ | 18.4$s$ | 12.1$s$ | 1.3$s$ | 12.0$s$ |
| ALBERT$_{\text{BASE}}$ + GFSA | 11M | 1.2$s$ | 1.7$s$ | 1.4$s$ | 62.1$s$ | 2.3$s$ | 19.7$s$ | 13.1$s$ | 1.4$s$ | 13.0$s$ |
| RoBERTa$_{\text{BASE}}$ [44] | 125M | 1.0$s$ | 1.4$s$ | 1.1$s$ | 47.0$s$ | 1.9$s$ | 15.0$s$ | 9.9$s$ | 1.2$s$ | 10.0$s$ |
| RoBERTa$_{\text{BASE}}$ + GFSA | 125M | 1.1$s$ | 1.4$s$ | 1.2$s$ | 50.4$s$ | 2.0$s$ | 16.3$s$ | 10.8$s$ | 1.2$s$ | 11.0$s$ |

Table 30: Inference time on causal language modeling tasks.

| Method | #Params | PTB | WikiText-2 | WikiText-103 | **Avg** |
|---|---|---|---|---|---|
| GPT2 [61] | 117M | 3.2$s$ | 7.4$s$ | 7.4$s$ | 6.0$s$ |
| GPT2 + GFSA | 117M | 5.5$s$ | 12.9$s$ | 12.9$s$ | 10.4$s$ |

Table 31: Inference time on ImageNet-1k

| Backbone | Method | #Layers | Inference Time |
|---|---|---|---|
| DeiT | DeiT-S | 12 | 52$s$ |
| | DeiT-S + GFSA | 12 | 53$s$ |
| | DeiT-S | 24 | 68$s$ |
| | DeiT-S + GFSA | 24 | 69$s$ |
| CaiT | CaiT-S | 24 | 105$s$ |
| | CaiT-S + GFSA | 24 | 107$s$ |
| Swin | Swin-S | 24 | 17$s$ |
| | Swin-S + GFSA | 24 | 17$s$ |

Table 32: Inference time on graph-level tasks

| Method | ZINC | PCQM4M | PCQM4Mv2 |
|---|---|---|---|
| Graphormer [94] | $8s$ | $99s$ | $31s$ |
| Graphormer + GFSA | $8s$ | $117s$ | $29s$ |

Table 33: Inference time on LibriSpeech datasets

| Method | LibriSpeech 100h | LibriSpeech 960h |
|---|---|---|
| Transformer | $328.1s$ | $323.7s$ |
| Transformer + GFSA | $329.5s$ | $343.3s$ |
| Branchformer [59] | $299.4s$ | $328.7s$ |
| Branchformer + GFSA | $305.5s$ | $354.1s$ |

Table 34: Inference time on the code defect prediction task

| Method | Inference Time |
|---|---|
| RoBERTa [44] | $22.4s$ |
| RoBERTa + GFSA | $23.9s$ |
| CodeBERT [23] | $23.8s$ |
| CodeBERT + GFSA | $24.1s$ |
| PLBART [2] | $37.7s$ |
| PLBART + GFSA | $39.3s$ |
| CodeT5-small [84] | $78.2s$ |
| CodeT5-small + GFSA | $82.5s$ |
| CodeT5-base [84] | $83.2s$ |
| CodeT5-base + GFSA | $88.5s$ |

# T    Results of the Strategy for Efficiency

In Tables 35 to 39, the results show that this strategy can reduce the increase in runtime and maintain performance compared to using GFSA for all layers.

Table 35: Comparison of performance using GFSA on all layers vs. GFSA$_{even}$ on even layers for GLUE tasks

| Datasets | #Params | CoLA | SST-2 | MRPC | QQP | STS-B | MNLI-m/mm | QNLI | RTE | **Avg** |
|---|---|---|---|---|---|---|---|---|---|---|
| BERT$_{BASE}$ [16] | 110M | 56.79 | 93.81 | 88.70 | 88.32 | 88.16 | 84.96/84.15 | 91.63 | 66.06 | 82.51 |
| BERT$_{BASE}$ + GFSA | 110M | 59.56 | 94.15 | 90.60 | 88.46 | 88.33 | 85.12/85.06 | 91.95 | 68.95 | 83.58 |
| BERT$_{BASE}$ + GFSA$_{even}$ | 110M | 58.80 | 93.69 | 90.50 | 88.47 | 88.27 | 85.13/85.02 | 91.65 | 70.76 | 83.59 |

Table 36: Comparison of training time (seconds per epoch) using GFSA on all layers vs. GFSA$_{even}$ on even layers for GLUE tasks

| Datasets | #Params | CoLA | SST-2 | MRPC | QQP | STS-B | MNLI-m/mm | QNLI | RTE | **Avg** |
|---|---|---|---|---|---|---|---|---|---|---|
| BERT$_{BASE}$ [16] | 110M | 17$s$ | 182$s$ | 17$s$ | 1483$s$ | 24$s$ | 2004$s$ | 580$s$ | 18$s$ | 541$s$ |
| BERT$_{BASE}$ + GFSA | 110M | 19$s$ | 192$s$ | 19$s$ | 1571$s$ | 25$s$ | 2147$s$ | 621$s$ | 20$s$ | 577$s$ |
| BERT$_{BASE}$ + GFSA$_{even}$ | 110M | 17$s$ | 185$s$ | 18$s$ | 1506$s$ | 24$s$ | 2061$s$ | 595$s$ | 18$s$ | 553$s$ |

Table 37: Comparison of performance using GFSA on all layers vs. GFSA$_{even}$ on even layers for causal language modeling tasks

| Method | #Params | PTB | WikiText-2 | WikiText-103 | **Avg** |
|---|---|---|---|---|---|
| GPT2 [61] | 117M | 19.513 | 20.966 | 15.939 | 18.806 |
| GPT2 + GFSA | 117M | 19.450 | 20.923 | 15.919 | 18.764 |
| GPT2 + GFSA$_{even}$ | 117M | 19.453 | 20.926 | 15.923 | 18.767 |

Table 38: Comparison of training time (seconds per epoch) using GFSA on all layers vs. GFSA$_{even}$ on even layers for causal language modeling tasks

| Method | #Params | PTB | WikiText-2 | WikiText-103 | **Avg** |
|---|---|---|---|---|---|
| GPT2 [61] | 117M | 89.1$s$ | 195.7$s$ | 9638.4$s$ | 3307.8$s$ |
| GPT2 + GFSA | 117M | 160.3$s$ | 354.2$s$ | 17424.6$s$ | 5979.7$s$ |
| GPT2 + GFSA$_{even}$ | 117M | 127.4$s$ | 279.1$s$ | 13761.4$s$ | 4722.6$s$ |

Table 39: Comparison of using GFSA on all layers vs. GFSA$_{even}$ on even layers for ImageNet-1k

| Method | Input Size | #Layers | #Params | Top-1 Acc | Runtime |
|---|---|---|---|---|---|
| DeiT-S | 224 | 12 | 43M | 79.8 | 551$s$ |
| DeiT-S + GFSA | 224 | 12 | 43M | 81.1 | 814$s$ |
| DeiT-S + GFSA$_{even}$ | 224 | 12 | 43M | 81.0 | 595$s$ |

# U GFSA in Linear Transformers

Table 40 shows the accuracy, runtime, and GPU usage results on Long Range Arena benchmark using ListOps and Image datasets.

Table 40: Comparison of accuracy (%), runtime ($s$ per 1,000 steps) and GPU usage (GB) on Long Range Arena benchmark

| Method | ListOps (2K) | | | Image (4K) | | |
|---|---|---|---|---|---|---|
| | Accuracy | Runtime | GPU usage | Accuracy | Runtime | GPU usage |
| Transformer [77] | 37.1 | 198.3 | 5.50 | 38.2 | 345.1 | 5.88 |
| Transformer+GFSA | **37.6** | 635.8 | 10.87 | **40.2** | 737.2 | 11.20 |
| Linformer [81] | 37.3 | 63.4 | 1.73 | 37.8 | 158.5 | 3.45 |
| YOSO-E [97] | 37.3 | 85.7 | 0.37 | 39.8 | 114.2 | 1.42 |
| Efficient Attention [70] | 36.9 | 49.2 | 0.57 | 40.2 | 121.1 | 1.14 |
| Efficient Attention + GFSA | **37.9** | 53.8 | 0.67 | **40.4** | 135.8 | 1.33 |

