# OpenReview forum: "Graph Convolutions Enrich the Self-Attention in Transformers!"
_NeurIPS.cc/2024/Conference — NeurIPS 2024 poster_

### Official Review · Reviewer_bzWT · 2024-07-09

**Soundness:** 3
**Presentation:** 3
**Contribution:** 1
**Rating:** 5
**Confidence:** 4

**Summary:**

To overcome the oversmoothing effects in general transformer settings, the author considers to treat the attention module as the graph filter and propose to use a polynomial graph filter (graph signal processing) techniques to alleviate oversmoothing. Specifically,  the author considers the induced attention matrix from features as the adjacency matrix and construct a filter that allows exhibiting high-pass and low-pass filter. As low-pass filters provides smoothing tendency and high-pass filter provides edge tendency, the author leverages an adaptive filter to alleviate oversmoothing effects on Transformer domain.

**Strengths:**

1. The idea is clear and well understood, the author leverages the idea in GNN fields to the transformer models.
2. The high-pass and low-pass filter is analyzed and the author used an approximation to approximate the polynomial filter.
3. The experiments covers many different applications of transformers.

**Weaknesses:**

1. The novelty is very limited. The idea of building adaptive high-pass, low-pass filter is well studied in GNN field and has been leveraged in many works to solve oversmoothing in GNN. The idea of attention module as graph filter is also proposed in many Graph transformer models. The main contribution of this paper is simply change the direction from using attention as graph filters to use graph filters as attentions, which is not very interesting.

**Questions:**

In the Theorem F.1, the description is not very clear. I assume  "the sum w0 +w1σi +wkσi^{k}  will be smaller than g(σi)" indicates that g(σi) will become smaller as i increase, but this depends on the scale of w1 and wk, even as σi^{k}->0 with i is large, if the rate wk/w1 is large, such tendency can only be guaranteed asymptotically, in the limited i case, this doesn't always hold true. Please correct me if I am wrong.

---

> ### Author Rebuttal · Authors · 2024-08-07
>
> We sincerely thank Reviewer bzWT for the review and feedback, highlighting our strengths in
> 1. Clear idea applying GNN concepts to transformer models.
> 2. Analytical approach to high-pass and low-pass filters with polynomial approximation.
> 3. Extensive experiments across various transformer applications.
>
> Below, we would like to address each of your questions and concerns:
>
> **Response to W1: comparison with GNN and Graph Transformer**
>
> While inspired by graph signal processing (GSP), we believe our contribution has significant novelty:
>
> GFSA uniquely interprets dynamic self-attention as a directed graph shift operator, unlike conventional GNNs and graph Transformers that typically deal with static undirected adjacency matrices. This difference sets GFSA apart from works like Polyformer [1] and Specformer [2].
>
> Polyformer combines a Graph Transformer with a graph filter and defines the graph filter using the adjacency matrix as a graph shift operator. It uses Transformers to learn the coefficients of polynomial expansion. Specformer uses the eigenvalues of the adjacency matrix as input to the Transformer to design a spectral GNN, but it doesn't alter the fundamental structure of Transformers, using them as encoder-decoders.
>
> Our approach is based on the following equation:
>
> $$\mathbf{y} = \mathbf{H}\mathbf{x} =  \begin{cases}{\color{blue}{\sum_{k=0}^{K} w_k \mathbf{\bar{A}}^k\mathbf{x}}} = {\color{red}{\mathbf{V}^\intercal \big(\sum_{k=0}^{K} w_k\mathbf{\Lambda}^k \big) \mathbf{V}\mathbf{x}}} & \text{if} \ \mathbf{\bar{A}} \text{ is undirected} \\\\ \color{blue}{\sum_{k=0}^{K} w_k \mathbf{\bar{A}}^k\mathbf{x}} & \text{if} \ \mathbf{\bar{A}} \text{ is directed} \end{cases}$$
>
> When implementing graph filters, the red spectral decomposition requires diagonalization of the graph shift operator matrix. In symmetric cases, such as undirected graphs, spectral decomposition is always guaranteed, allowing consistent application of the red method. However, in directed graphs, spectral decomposition is not guaranteed and thus cannot always be applied.
>
> In contrast, the blue matrix-polynomial method does not require diagonalization. This enables consistent application of graph signal processing to both directed and undirected matrices. This approach is essential for self-attention matrices where diagonalization is impossible, making GFSA more broadly applicable and theoretically sound.
>
>  Furthermore, we propose a method to approximate higher-order polynomial terms. This approximation is key to achieving performance gains without computational costs.
>
> In conclusion, GFSA integrates GSP with Transformers, addressing non-diagonalizable self-attention matrices and offering a scalable model applicable across various domains.
>
> > [1] Bo et al., "Specformer: Spectral Graph Neural Networks Meet Transformers." ICLR 2023
> >
> > [2] Jiahong et al., “PolyFormer: Scalable Node-wise Filters via Polynomial Graph Transformer”, KDD 2024
>
>
> **Response to Q1: Theorem 3.1**
> Thanks for your question. We have corrected the typo $g(\sigma_i) \rightarrow \sigma_i$ on line 696.
>
> We agree that if the ratio $w_k/w_1$ is large, it may not always be a low-pass filter. Theorem 3.1 proves that under specific coefficient conditions, the graph filter becomes either low-pass or high-pass. For example, if the coefficients are all positive and their sum is 1, the graph filter becomes a low-pass filter. Alternatively, if the coefficients are $w_k=(-\alpha)^k$ with sufficient large $K$, the graph filter becomes a high-pass filter.
>
> Additionally, we have revised Theorem 3.1 to address your concern:
>
> **(Theorem 3.1 Filter characteristics based on coefficient values)**. Let $\bar{\mathbf{A}}$ be a self-attention matrix interpreted as a graph with connected components. Consider the polynomial graph filter defined by $\sum_{k=0}^K w_k \bar{\mathbf{A}}^k$, where $w_2, w_3, \ldots, w_{K-1} = 0$ and only $w_0$, $w_1$, and $w_K$ are non-zero. If the coefficients $w_k$ for $k=0,1,K$ are positive and their sum is 1, then the polynomial filter acts as a low-pass filter, attenuating high-frequency components and promoting smoothness across the graph. Conversely, if $w_k=(-\alpha)^k$ for $k=0,1,K$ and $\alpha \in (0,1)$, the polynomial filter exhibits high-pass filter behavior.
>
> **Proof.** For the case of low-pass filter, the graph filter acts as low-pass filter if $|g(\sigma_i)/g(\sigma_1)|<1$ for $\forall i \geq 2$ where $\sigma_i$ indicates the $i$-th singular value of $\bar{\mathbf{A}}$ . From the assumption that the coefficients $w_k$ for $k=0,1,K$ are positive and their sum is 1, we derive that
>
> $\left|g(\sigma_1)\right| = \left|w_0 + w_1 + w_K\right| = 1$,
>
> and
>
> $\left| g(\sigma_i)\right| =\left| w_0 + w_1\sigma_i  + w_K\sigma_i^k \right| < \left| w_0 + w_1\sigma_i  + w_K\sigma_i \right| < \left| w_0 + w_1 + w_K \right| = 1$.
>
> Therefore, as $|g(\sigma_i)/g(\sigma_1)|<1$, the graph filter acts as low-pass filter.
>
> For the case of high-pass filter, the graph filter acts as a high-pass filter if $|g(\sigma_i)/g(\sigma_1)| > 1$ for $\forall i \geq 2$. From the assumption that the coefficients $w_k=(-\alpha)^k$ for $k=0,1,K$ and $\alpha \in (0,1)$ with sufficient large $K$, we derive that
>
> $\left| \frac{\lim_{K \rightarrow \infty} g(\sigma_i)}{\lim_{K \rightarrow \infty} g(\sigma_1)} \right|  = \left|\frac{\lim_{K \rightarrow \infty} w_0 + w_1 \sigma_i + w_K \sigma_i^K}{\lim_{K \rightarrow \infty} w_0 + w_1  + w_K} \right|  =\left|\frac{\lim_{K \rightarrow \infty} -\alpha + \alpha^2 \sigma_i^2 + (-\alpha)^K \sigma_i^K}{\lim_{K \rightarrow \infty} -\alpha + \alpha^2 + (-\alpha)^K}\right| = \left|  \frac{ -\alpha + \alpha^2 \sigma_i^2}{ -\alpha + \alpha^2 }\right|   = \left| \frac{ -1 + \alpha\sigma_i^2}{ -1 + \alpha }\right|  > 1$
>
> Therefore, as $|g(\sigma_i)/g(\sigma_1)| >1$ with sufficient large $K$, the graph filter acts as high-pass filter.
>
> This proof supports that the behavior filter depends directly on the sign and values of the coefficients.

---

> > ### Comment · Reviewer_bzWT · 2024-08-09
> >
> > Thanks for the clarification on the matrix power part. While I believe that such matrix power has been utilized commonly in the early-works of GNN resolving heterophilic graphs (such as [1]), a taylor approximation does provide an easy computation savoir. However, since the self-attention matrix is dynamically changed rather than a static adjacency, this adds on a matrix power computation for each forward phase, which I believe is non-trivial computation cost. I also believe that this part should be added to the main text rather than appendix, this confused the reader's understanding of what the paper is trying to achieve.
> >
> > Since the paper proposed to use matrix power directly as a filter for both high and low pass filter learning, it becomes important that Theorem 3.1 should be carefully inspected as it lays the theoretical foundation for the paper. For your revised theorem and proof, I have the following questions:
> > 1. The proof for your high-pass filter case doesn't look right. Although the conclusion I think remains, but in the third step, isn't that be $\frac{1-\alpha \sigma_i + (- \alpha)^K \sigma_i^K}{1 - \alpha+(- \alpha)^K} $ ?
> > 2. The assumption for low pass filter requires learnable constrained by sum to 1 and the assumption for high-pass filter is even stricter. In general, what constrains can we make in either loss or forward process to ensure such conditions? What guarantees we can learn such weights that satisfy the assumptions?
> > 3. Theorem 3.1 is applied to the exact A_{k} case, while the author has provided a bound for how the approximated polynomial is close to the exact case, it seems non-trivial to directly adapt the conclusion of exact singular value analysis to the approximate case. Especially the assumption requires K to be sufficiently large. I made a simple derivation following the revised proof with the formula Equation 10 replacing the A-{K} polynomial. From what I observed, the final ratio is:
> >
> > $|g(\sigma_i)/g(\sigma_1)| = \frac{\sigma (1- \sigma)}{1-\frac{K-1}{K-2}}$ with $K->\infty$. This suggests that the final filter depends only on the singular value (or the learned directed normalized adjacency matrix), but independent of the learnable w_{0},w_{1},w_{2}. If this is the case, then I would question the necessity of the learnable parameters as this is similar to the red case which is altering the eigenvalue of the learned A directly.
> >
> > I am very concerned about the accuracy of this part as this is a key novelty (or at least theoretically) and the key reason why method should work and I am worried that the non-trival adaptation from exact case to approximate case leads to a illusion that $w_0,w_1, w_k$ is the key.
> >
> >
> >
> > [1] Abu-El-Haija, S., Perozzi, B., Kapoor, A., Alipourfard, N., Lerman, K., Harutyunyan, H., Steeg, G.V. &amp; Galstyan, A.. (2019). MixHop: Higher-Order Graph Convolutional Architectures via Sparsified Neighborhood Mixing. ICLR.

---

> ### Author Response · Authors · 2024-08-11
>
> Dear Reviewer bzWT,
>
> Thank you for your comments on the computational complexity of GFSA and your careful review and follow-up questions of Theorem 3.1 and its proof. We want to address your questions and concerns in two main parts.
>
> ---
>
> **[Response to the computational complexity of GFSA]**
>
> First, to address the computational overhead of GFSA, we have implemented a selective application strategy, as detailed in *Section 6 of* *our main text*. By applying GFSA only to even-numbered layers, we effectively reduce runtime increases while maintaining comparable performance to full-layer integration.
>
> Moreover, we understand your concern about the added complexity in the context of Transformers. As you know, the original self-attention requires quadratic computational cost, which led to the development of efficient/linear Transformers with linear complexity in recent years. To address this, we want to highlight the compatibility of GFSA with efficient/linear Transformers.
>
> As detailed in our response to `Reviewer k2id`, we have conducted experiments showing that:
>
> 1. GFSA can be effectively integrated with Efficient attention [1], one of the linear Transformers, maintaining its efficiency benefits while improving performance. Here are some key results:
>
> |  | ListOps (2K) | Image (1K) |
> | --- | --- | --- |
> | Efficient Attention | 36.9% / 49.2s / 0.57GB | 40.2% / 121.1s / 1.14GB |
> | **Efficient Attention + GFSA** | **37.9%** / 53.8s / 0.67GB | **40.4%** / 135.8s / 1.33GB |
>
> > Note: Results are shown as Accuracy (%) / Running time (s/1K-steps) / Peak training memory usage (GB)
>
> 2. There are potential efficiency improvements for GFSA itself. By designing $\mathbf{\bar{A}}$ as $\mathbf{\bar{A}}=\mathbf{QK}$, we can compute $\mathbf{\bar{A}}^2\mathbf{V} = (\mathbf{QK})(\mathbf{QK})\mathbf{V}$ in a manner similar to Efficient attention mechanism, maintaining linear complexity even for the squared operation in GFSA.
>
> These findings show that GFSA with a linear Transformer can enhance the performance while preserving the efficiency benefit. For a more detailed discussion of GFSA compatibility with a linear/efficient Transformer, see our response to `Reviewer k2id`, "**4. Limitations (scalability and extendability)**".
>
> > [1] Shen, Zhuoran, et al. "Efficient attention: Attention with linear complexities." WACV 2021.

---

> ### Author Response · Authors · 2024-08-11
>
> **[Responses to Theorem 3.1 and its proof]**
>
> **Response to Q1.**
>
> You are correct, and we thank you for pointing this out. There was a typo in the third step of the high-pass filter proof. We apologize for any confusion this may have caused. To clarify, the corrected one is as follows:
>
> $\left| \frac{\lim_{K \rightarrow \infty} g(\sigma_i)}{\lim_{K \rightarrow \infty} g(\sigma_1)} \right|  = \left|\frac{\lim_{K \rightarrow \infty} w_0 + w_1 \sigma_i + w_K \sigma_i^K}{\lim_{K \rightarrow \infty} w_0 + w_1  + w_K} \right|  =\left|\frac{\lim_{K \rightarrow \infty} 1 - \alpha \sigma_i + (-\alpha)^K \sigma_i^K}{\lim_{K \rightarrow \infty} 1 - \alpha + (-\alpha)^K}\right| = \left|  \frac{ 1 -\alpha\sigma_i }{ 1 -\alpha }\right|  > 1$
>
>
> **Response to Q2.**
>
> As you mentioned, Theorem 3.1 assumes that the coefficients have specific values, and the coefficient of the graph filter learned through deep learning cannot always satisfy those exact values. However, what we want to clarify through this theorem is that **the value of coefficients determines the characteristics of the graph filter.**
>
> Our goal in learning the coefficients of GFSA is not to create an exact low-pass or high-pass filter, but to design a graph filter that can appropriately employ frequency information of various scales for downstream tasks. The theorem provides a theoretical foundation for understanding how the coefficients influence the behavior of the filter. In practice, we do not enforce strict constraints on the coefficients during training. Instead, we allow the model to learn the most appropriate coefficients for the task at hand, which may result in filters that combine both low-pass and high-pass characteristics to varying degrees.
>
>
> **Response to Q3.**
>
> Theorem 3.1 is indeed a proof for the exact $\mathbf{\bar{A}}^K$. However, contrary to your concern, **the characteristic of our GFSA still depends on the value of coefficients** $w_k$. To address your concerns, we extend the proof using the approximated $\mathbf{\bar{A}}^K$ used by our GFSA.
>
> **Proof for low-pass filter:** We prove the low-pass filter result for GFSA. For the case where $w_0, w_1$, and $w_K$ are positive and their sum is 1, we prove that $\left| g(\sigma_i) \right| < 1$.
>
> Since $\sigma_1 = 1$, we have:
>
> $g(\sigma_1)=w_0+w_1\sigma_1+w_K(\sigma_1+(K-1)(\sigma_1^2-\sigma_1))=w_0+w_1+w_K=1$.
>
> Hence, proving the low-pass filter result is equivalent to showing $\left| g(\sigma_i) \right| < 1$, and $g(\sigma_i)$ is bounded as follows:
>
> $\left| g(\sigma_i)\right| =\left| w_0 + w_1\sigma_i  + w_K(\sigma_i + (K-1) (\sigma_i^2-\sigma_i)) \right| \overset{(1)}{<} \left| w_0 + w_1\sigma_i  + w_K\sigma_i \right| = \left|\sigma_i \right|< 1$
>
> This inequality (1) satisfies since $\left| \sigma_i + (K-1) (\sigma_i^2-\sigma_i) \right| = \left| \sigma_i((K-1)\sigma_i - (K-2)) \right| < \left|\sigma_i((K-1)-(K-2))\right| = \sigma_i$.
>
> Therefore, we complete the proof for the low-pass filter.
>
> **Proof for high-pass filter:** When $w_k = (−\alpha)^k/(k+1)$ where $k = 0, 1, K$ and $\alpha \in (0,1)$, we prove:
>
> $\left| \frac{\lim_{K \rightarrow \infty} g(\sigma_i)}{\lim_{K \rightarrow \infty} g(\sigma_1)} \right| = \left|\frac{\lim_{K \rightarrow \infty} w_0 + w_1 \sigma_i + w_K (\sigma_i + (K-1) (\sigma_i^2-\sigma_i))}{\lim_{K \rightarrow \infty} w_0 + w_1  + w_K (1 +  (K-1) (1-1))} \right| \\ =\left|\frac{\lim_{K \rightarrow \infty} 1 - \frac{\alpha}{2}\sigma_i + \frac{(-\alpha)^K}{(K+1)}(\sigma_i + (K-1) (\sigma_i^2-\sigma_i))}{\lim_{K \rightarrow \infty} 1 - \frac{\alpha}{2} + \frac{(-\alpha)^K}{(K+1)}}\right|$
> $= \left|\frac{1 - \frac{\alpha}{2}\sigma_i + \big(\lim_{K \rightarrow \infty}\frac{(-\alpha)^K}{(K+1)}(\sigma_i + (K-1) (\sigma_i^2-\sigma_i))\big)}{1 - \frac{\alpha}{2}}\right| \\ \overset{(1)}{=}\left|\frac{1 - \frac{\alpha}{2}\sigma_i}{1 - \frac{\alpha}{2}}\right| > 1$
>
> This equality (1) satisfies since  $\lim_{K \rightarrow \infty}\frac{(-\alpha)^K}{(K+1)}(\sigma_i + (K-1) (\sigma_i^2-\sigma_i)) = \lim_{K \rightarrow \infty}\frac{(-\alpha)^K}{(K+1)}((K-1) (\sigma_i^2-\sigma_i)) = \lim_{K \rightarrow \infty}(-\alpha)^K\frac{(K-1)}{(K+1)}(\sigma_i^2-\sigma_i) = 0$.
>
> Therefore, it shows that the graph filter acts as a high-pass filter.
>
> ---
>
> We appreciate your careful analysis of our work. You provide an interesting perspective, which we will gladly add to the revised paper. Your comments have allowed us to focus Theorem 3.1 more closely on our GFSA, which has significantly improved our theoretical foundation.
>
> However, according to our proof, the final ratio you derived is difficult for us to understand. We have encountered some difficulty as the final ratio has $K$ and seems to diverge when $K$ approaches infinity. *If our response does not address your concerns, could you kindly provide us with your derivation?*

---

> ### Comment · Reviewer_bzWT · 2024-08-12
>
> $\left| \sigma_i + (K-1) (\sigma_i^2-\sigma_i) \right| = \left| \sigma_i((K-1)\sigma_i - (K-2)) \right| < \left|\sigma_i((K-1)-(K-2))\right| = \sigma_i$ looks suspicious. When $K->\infty$, I don't think you can make an inequality of shrinking $\sigma_i(K-1)$ to $(K-1)$, this is like claiming 2 $\infty$ < 3 $\infty$, which doesn't make sense. Plus, it is weird to see something that tends to $\infty$ can be bounded by 1.
>
> So I don't think the low-pass filter proof is correct. In fact, It think it means that w2 and $\sigma_i$ term dominates the result. If you divide the K-1 to both denominator and numerator, w0, w1 doesn't seem to have any effect on the final convergence.

---

> > ### Author Response · Authors · 2024-08-13
> >
> > Dear Reviewer bzWT,
> >
> > Thanks for your thoughtful review.
> >
> > However, contrary to your concern, the proof of the low-pass filter does not assume that $K$ approaches infinity. It only assumes that the coefficients are positive and their sum is 1, which is why the inequality holds. The assumption that $K$ goes to infinity is only applied in the proof of the high-pass filter.
> >
> > We hope our response addresses your concern.

---

> > > ### Comment · Reviewer_bzWT · 2024-08-13
> > >
> > > Thanks for pointing that out. I get the point for the low-pass filter now. But then doesn't this implies that the choice of K directly affects if it is a low-pass filter or high-pass? Honestly, although the theorem might be correct after these many conditions, it starts to become far from reality from my perspective. Ideally, I think this should be only related to the sign of w0,w1, and w2, but then things become rather complex and even involve with the choice of K. So how exactly can a model learn some layers of high pass filter and some layers of low pass filter theoretically? When does this occur? Your added sensitivity plot suggests that K doesn't affect the model performance very much. What does that indicate in terms of the fixed theoretical analysis?

---

> > > > ### Author Response · Authors · 2024-08-13
> > > >
> > > > Dear Reviewer bzWT,
> > > >
> > > > We appreciate your questions regarding the role of $K$ and the nature of our graph filter. We address your questions below:
> > > >
> > > > **[Learning high-pass and low-pass filters]**
> > > >
> > > > Our goal is not to create theoretically exact high-pass or low-pass filters. Instead, we aim to learn optimal graph filters based on coefficients, improving the original self-attention.
> > > >
> > > > This adaptive behavior can be found in the filter response visualizations provided in Figure 2 of our one-page PDF in general response. We visualized the filters for all 12 layers of BERT with and without GFSA applied. In Figure 2, GFSA learns diverse filter types across layers, evolving from low-pass in early layers to a mix of low and high-pass in middle layers and shifting towards higher-frequency responses in deeper layers. Each layer does not function as a theoretically exact high-pass or low-pass filter but rather as a graph filter that can use frequency information of various scales to suit the downstream task via learnable coefficients.
> > > >
> > > > From the perspective of using learnable coefficients, this approach is similar to polynomial GNNs, such as GPR-GNN [1] and BernNet [3]. They can learn arbitrary filters instead of fixed filters by learning $w_k$. For example, in the case of GPR-GNN [1], any graph filter can be expressed by applying gradient descent to the polynomial coefficients.
> > > >
> > > > **[The role of $K$ in GFSA]**
> > > >
> > > > Thank you for your question about the choice of $K$.
> > > >
> > > > Our goal is not to construct an exact low/high pass filter but to create a graph filter that improves on self-attention. We add an approximated $\mathbf{\bar{A}}^K$ to achieve maximum effectiveness with minimum overhead.
> > > >
> > > > In the case of Theorem 3.1 applied to GFSA, the result of the proof does not change depending on the value of $K$ defined as a hyper-parameter. In other words, $K$ does not directly affect whether the graph filter becomes a high-pass or low-pass filter.
> > > >
> > > > Of course, in the proof of the high-pass filter, there is a condition that $K$ has a sufficiently large value. However, since the characteristic of the graph filter is determined by the coefficient once the value of $K$ is determined, this condition does not mean that $K$ directly affects the characteristic of the GFSA graph filter.
> > > >
> > > > As you observed in the sensitivity study, the performance may not change drastically depending on the value of $K$ itself. This is because the learned coefficients have a more impact on filter behavior than the choice of $K$. Also, Figure 2 in a one-page PDF supports this, showing the varied filter response for each layer for a fixed $K$. Therefore, we can see that the coefficients affect the characteristics of the GFSA graph filter.
> > > >
> > > > > [1] Chien, Eli, et al. "Adaptive universal generalized pagerank graph neural network." *arXiv preprint arXiv:2006.07988*(2020).
> > > > >
> > > > > [2] Defferrard, Michaël, Xavier Bresson, and Pierre Vandergheynst. "Convolutional neural networks on graphs with fast localized spectral filtering." NeurIPS, 2016.
> > > > >
> > > > > [3] Mingguo He, Zhewei Wei, and Hongteng Xu. "Bernnet: Learning arbitrary graph spectral filters via Bernstein approximation." NeurIPS, 2021.
> > > >
> > > > ---
> > > >
> > > > We hope this clarification addresses your concerns.
> > > >
> > > > Best regards,
> > > >
> > > > GFSA authors

---

> > > > > ### Comment · Reviewer_bzWT · 2024-08-13
> > > > >
> > > > > Thanks for the author's detailed response. I have decided to raise my score to 5. While I agree that the experimental results show positive evidence on the learning of high-pass filter, it is still not clear to me how the revised theory really connect to these results. I appreciate these useful discussions.

---

> > > > > > ### Author Response · Authors · 2024-08-14
> > > > > >
> > > > > > Dear Reviewer bzWT,
> > > > > >
> > > > > > It's really encouraging to see that our response effectively addresses the concerns you raised. Thank you for recognizing the efforts we've put into our work.
> > > > > >
> > > > > > To show  how our extended Theorem 3.1 with GFSA can be connected to constructing the type of filter in practice, we use a toy example to help you understand that
> > > > > > 1) we show that the coefficients determine the filter type and
> > > > > > 2) we show that the role of K does not change the filter type.
> > > > > >
> > > > > > **[Filter type according to cofficients]**
> > > > > >
> > > > > > We assume the singular values $\sigma_1, \cdots ,\sigma_5$ as $\\{1.0, 0.8, 0.6, 0.4, 0.2 \\}$, and $K$ is 5. The graph filter of GFSA is defined as $g(\sigma_i)=w_0 + w_1\sigma_i + w_K(\sigma_i + (K-1)(\sigma_i^2-\sigma_i))$.
> > > > > >
> > > > > > In the proof of extended Theorem 3.1 with GFSA, we can see that if we use positive coefficients that sum to 1, it is consistent with our proof that it is an exact low-pass filter because $g(\sigma_i)$ for $i > 1$ is always less than 1.
> > > > > >
> > > > > > For the exact high-pass filter, assuming the coefficients to be $w_1, w_2, w_K = (-0.5)^0 /1, (-0.5)^1 /2, (-0.5)^K / (K+1)$ as per the proof, we can see that the results in Table below are consistent with the ratio $\frac{g(\sigma_i)}{g(\sigma_1)} > 1$.
> > > > > >
> > > > > > And if we set the coefficients to arbitrary values like the last row, we can create an arbitrary filter.
> > > > > >
> > > > > > That is, the characteristics of the filter change depending on the coefficients, and GFSA can learn any graph filter that fits the downstream task by learning the coefficients without strict restrictions.
> > > > > >
> > > > > > | filter type | coefficients ($w_0, w_1, w_K$) | $ g(\sigma_1), \cdots ,g(\sigma_5)$ |
> > > > > > | --- | --- | --- |
> > > > > > | exact low-pass filter | (0.2, 0.4, 0.4) | 1.00, 0.58, 0.30, 0.14, 0.10 |
> > > > > > | exact high-pass filter | $((-0.5)^0 / (1), (-0.5)^1 / (2), (-0.5)^K / (K+1))$ | 0.75, 0.80, 0.85, 0.90, 0.95 |
> > > > > > | arbitrary filter | (-0.1 , 1.0 , -0.4) | 0.50, 0.64, 0.65, 0.52, 0.28 |
> > > > > >
> > > > > > **[The role of $K$ on filter type]**
> > > > > >
> > > > > > To understand the effect of $K$ on the characteristics of the graph filter, we analyzed the graph filter according to the value of $K$. For the exact high-pass filter, we report the filtered singular values according to the value of $K \in \\{3, 5, 10\\}$. As shown in the table below, even when the value of $K$ changes, the graph filter still maintains the characteristics of the exact high-pass filter.
> > > > > >
> > > > > > | $K$ | $ g(\sigma_1), \cdots ,g(\sigma_5)$ |
> > > > > > | --- | --- |
> > > > > > | 3 | 0.719, 0.785, 0.846, 0.902, 0.954 |
> > > > > > | 5 | 0.745, 0.799, 0.852, 0.903, 0.952 |
> > > > > > | 10 | 0.750, 0.800, 0.850, 0.900, 0.950 |
> > > > > >
> > > > > > ---
> > > > > >
> > > > > > We hope our explanation clears up your one last remaining concern.
> > > > > >
> > > > > > Best regards,
> > > > > >
> > > > > > GFSA Authors

---

### Official Review · Reviewer_k2id · 2024-07-11

**Soundness:** 1
**Presentation:** 3
**Contribution:** 2
**Rating:** 4
**Confidence:** 4

**Summary:**

This paper proposes a novel approach to enhance the self-attention mechanism in transformers by drawing on graph signal processing principles. The authors reframe the standard self-attention as a low-pass graph filter and introduce a more generalized filter (GFSA) capable of capturing low-pass, high-pass, or combined filtering behaviors. They evaluate the effectiveness of GFSA across diverse tasks, showcasing its potential to improve performance.

**Strengths:**

- The core idea is easily grasped and demonstrates promising results across various transformer variants in different domains.
- The authors evaluate GFSA on a wide range of tasks spanning natural language processing, computer vision, speech recognition, and graph regression, highlighting its versatility.

**Weaknesses:**

- While the oversmoothing problem motivates the proposed method, the experiments lack a focused analysis of its impact on transformers with varying depths (number of layers).
- While the paper conducted experiments in a variety of tasks in different fields to show the versatility of the proposed method, not study in depth each task could also be a potential weakness. In particular for the graph regression task, the authors only considered two datasets and a single baseline model (and the improvement over the larger model on PCQMv2 is marginal 0.860 vs 0.862). Stronger baselines such as [GPS](https://arxiv.org/abs/2205.12454) and additional datasets would provide a more rigorous assessment of GFSA's practical value. Additionally, the choice of the baselines in other tasks might also suffer from the same weakness, as I believe most of them are relatively old methods. Finally, their performance statement in Fig.1 may be misleading to practitioners without acknowledging the limitations with their selected baselines.
- Given the abundance of research on oversmoothing in transformers, the paper should include comparisons with more recent and relevant methods (e.g., [64]) to strengthen its claims. Currently, the paper only compares to very few comparison partners (ContraNorm [27] and [73]) or even without any comparison in some tasks.
- The paper states that the oversmoothing issue is due to the low-pass filter nature of the self-attention. By introducing a 2nd order term in Eq.10, they claim that their proposed method can learn both low-pass and high-pass filters, and even combined filters. To validate this empirically, a deeper analysis of the learned coefficients $w_0,w_1,w_K$ across different layers would provide concrete evidence for the GFSA's ability to learn diverse filter types.

**Questions:**

- L205: for the self-attention with the skip connection, shouldn't $w_0$ be positive?
- In the NLP experiments, did the pretrained models use the standard self-attention? If so, does this approach make sense to replace them with GFSA only during fine-tuning?
- The paper only discusses the runtime overheads. Does GFSA introduce any potential memory overheads?

**Limitations:**

The paper discussed the additional runtime overhead of GFSA compared to the original self-attention. However, other potential limitations are not discussed, such as potential memory overheads, scalability of GFSA to long sequences, or extendability to efficient/linear transformers.

---

> ### Author Rebuttal · Authors · 2024-08-07
>
> We sincerely thank Reviewer k2id for the review and feedback, highlighting our strengths in
> 1. Easily understandable core idea with promising results across diverse transformer variants.
> 2. Versatile performance demonstrated across a wide range of tasks in multiple domains.
>
> Below, we would like to address each of your questions and concerns:
>
> **Response to W1: varying depth**
>
> We would like to highlight the comprehensive layer-wise analysis we have conducted.
>
> Fig. 2 compares DeiT and DeiT+GFSA across all 24 layers. The cosine similarity plot (Fig. 2b) shows how GFSA mitigates oversmoothing as depth increases. DeiT shows increasing cosine similarity in deeper layers, indicating oversmoothing, while DeiT+GFSA maintains lower, more stable similarity across layers.
> Similar results are observed for BERT on STS-B (Fig. 3) and Graphormer on ZINC (Fig. 4).
> The singular value distributions (Fig. 2c, 3c, 4c) further support our findings.
>
> This layer-wise analysis provides strong evidence of GFSA's effectiveness in addressing oversmoothing across the full depth of the studied models.
>
> **Response to W2: weak baseline**
>
> To provide a rigorous assessment of GFSA's practical value, we have conducted additional experiments applying GFSA to SOTA models that use self-attention, GPS [1], and Graph-ViT [2] on LRGB [3] and Benchmarking GNNs [4]. For experimental results, we refer the reader to Table 1 and Table 2 in the PDF file of the general response.
>
> For GPS, we replace its self-attention module with our GFSA while maintaining its best configuration and other hyperparameters. For Graph-ViT, we apply GFSA to the Hadamard self-attention method. These experiments follow the settings used in both papers.
>
> Our new results show consistent improvements across various datasets. These additional experiments show the versatility and effectiveness of GFSA beyond our initial studies.
>
> > [1] Rampášek et al. "Recipe for a general, powerful, scalable graph transformer." NeurIPS 2022
> >
> > [2] He et al. "A generalization of vit/mlp-mixer to graphs." ICML 2023
> >
> > [3] Dwivedi et al. "Long range graph benchmark." Advances in NeurIPS 2022
> >
> > [4] Dwivedi et al. "Benchmarking graph neural networks." JMLR
>
> **Response to W3: comparison with other recent and relevant methods**
>
> We would like to highlight the analysis already present in our paper.
>
> As shown in Appendix Table 13, we have included comparisons with several SOTA methods. Notably, we have compared GFSA with NeuTRENO, a recent method designed to address oversmoothing. Our GFSA outperforms NeuTRENO while maintaining a similar parameter count to the original DeiT-S.
>
> Table 13 not only includes models addressing oversmoothing but also other recent advanced models such as SpectFormer and SVT. This provides a broader context for GFSA's performance across different types of SOTA approaches.
>
> Regarding [64], we were unable to include a direct comparison since the code is not public. However, we believe our current comparisons, which include both oversmoothing-specific methods and other recent innovations, provide a comprehensive evaluation of GFSA's effectiveness.
>
>
> **Response to W4: learned filter analysis**
>
> We have addressed your suggestion by visualizing the frequency responses, which directly represent the impact of learned coefficients, for all 12 layers of BERT with and without GFSA. These visualizations are now included in a PDF file of the general response.
>
> Our analysis reveals:
> 1. GFSA learns diverse filter types across layers, evolving from low-pass in early layers to a mix of low and high-pass in middle layers and shifting towards higher-frequency responses in deeper layers.
> 2. BERT+GFSA consistently shows higher magnitude responses at higher frequencies compared to vanilla BERT, especially in deeper layers.
> 3. While vanilla self-attention functions primarily as a low-pass filter, GFSA utilizes a wider range of frequencies.
>
> These findings provide evidence for GFSA's effectiveness in mitigating oversmoothing and preserving high-frequency information. We thank the reviewer for this valuable suggestion, which has enhanced the depth and rigor of our paper's contribution.
>
>
> **Response to Q1: skip-connection**
>
> We appreciate your careful observation regarding self-attention with skip connections. In standard Transformers, the residual term is positive. However, in our GFSA, $w_0$ serves a slightly different purpose. $w_0$ adjusts the weight of the identity matrix within each attention head, controlling how much of the original input should be preserved. Unlike traditional skip connections, $w_0$ can be learned, set to a fixed value, or even set to 0, depending on task requirements. When learned, $w_0$ provides flexibility to adapt the residual-like connection based on the task. When fixed to a positive value, it can more closely mimic traditional residual connections.
>
> **Response to Q2: fine-tuning PLM with different architecture**
>
> Thank you for your question about GFSA implementation in our NLP experiments.
>
> We replaced standard self-attention with GFSA during the fine-tuning phase of pre-trained models. This approach aligns with recent practices in the field, as seen in works that modify pre-trained model structures during fine-tuning [1,2,3].
>
> Our approach offers flexibility in coefficient initialization and learning. One option is to initialize $w_0$ and $w_K$ to 0 (see Appendix A), allowing them to be learned during fine-tuning. This strategy provides a smooth transition from standard self-attention to GFSA, enabling task-specific adaptation.
>
> > [1] Fine-tune BERT with Sparse Self-Attention Mechanism, EMNLP-IJCNLP 2019
> >
> > [2] Fast transformers with clustered attention, NeurIPS 2020
> >
> > [3] Enhancing Self-Attention with Knowledge-Assisted Attention Maps, ACL 2022
>
> **Response to Q3: memory overheads**
>
> Please refer to our **Response to Q1** of Reviewer XKs2.

---

> > ### Comment · Reviewer_k2id · 2024-08-08
> >
> > Thank you for your detailed response. However, some of my concerns are still not fully addressed.
> > - W1: to clarify my point, my concern is whether oversmoothing really correlates with the transformer's performance. So a more fair setup would be to measure the performance of a transformer model with varying depth and compare it with GFSA (for each layer number).
> > - W2: thank you for the additional results. However, the improvement is not statistically significant as is most of the time within the confidence interval of the baseline method.
> > - W3: thank you for pointing me to the relevant tables. I don't think putting important results in the Appendix is a good practice. Furthermore, after checking the results, I found only one additional baseline (DeiT-S + NeuTRENO) conducted on an image classification task, and the improvement is again very marginal. I believe a more systematic comparison of the proposed method and more recent baselines on all considered tasks would largely strengthen the conclusions of this work.
> > - Unfortunately, you did not address my points mentioned in the **Limitations**, particularly the scalability of GFSA to long sequences, or extendability to efficient/linear transformers. I believe these points are essential to assess the practical impact of this work.

---

> ### Author Response · Authors · 2024-08-11
>
> Dear Reviewer k2id,
>
> We would like to thank the reviewers for carefully analyzing our work and raising concerns that remain unresolved. We address the 4 concerns you raised below:
>
> ---
>
> **1. Regarding oversmoothing correlation with Transformer performance:**
>
> We would like to highlight that we have reported such an analysis, which provides evidence for the effectiveness in mitigating oversmoothing and improving performance across different depths.
>
> As shown in Table 14, while DeiT can potentially improve with 24 layers due to increased parameters, our cosine similarity analysis (See Fig.2 (a)) shows that oversmoothing at deeper depths could limit performance gains. This is where GFSA shows its effectiveness. GFSA allows the DeiT to better use the increased capacity of deeper layers. For example, GFSA improves DeiT-S by 2.54 points at 12 layers and 1.46 points at 24 layers. GFSA also shows larger improvements over ContraNorm at deeper depths.
>
> For convenience, we re-present the results of DeiT-S in Table 14 below:
>
> | #Layers | 12 | 16 | 24 |
> | --- | --- | --- | --- |
> | DeiT-S | 77.32 | 78.25 | 77.69 |
> | DeiT-S + ContraNorm | 77.80 | 79.04 | 78.67 |
> | **DeiT-S + GFSA** | **79.86** | **80.83** | **79.15** |
>
> This depth-wise analysis, combined with our previous layer-wise behavior studies (e.g., cosine similarity), provides our empirical results of GFSA in mitigating oversmoothing and improving performance in varying model depths.
>
> ---
>
> **2. Statistical significance of improvements:**
>
> While the improvements in CIFAR-10 (p = 0.184, paired t-test comparing GraphGPS and GraphGPS+GFSA) and MNIST (p = 0.108, paired t-test) may not be statistically significant, Cohen's D values of 0.455 and 0.565 respectively indicate medium effect sizes. These medium effect sizes suggest that ***our improvements are still meaningful*** in practice. As [1] argues, effect size can reveal practical significance that p-values alone might miss
> Note that statistical testing was only feasible with GraphGPS, which provided results from 10 runs, unlike Graph-ViT.
>
> For Graph-ViT, it is significant that GFSA can support Graph-ViT to outperform the Graph-MLP-Mixer on the MolTOX21 dataset, as shown below.
>
> | Method (GINE as a graph encoder) | MolTOX21 |
> | --- | --- |
> | Graph-MLP-Mixer | 0.7868 ± 0.0043 |
> | Graph-ViT | 0.7851 ± 0.0077 |
> | **Graph-ViT + GFSA** | **0.7895 ± 0.0069** |
>
> In this context, we would appreciate it if you could understand that our GFSA can be effective in broader graph benchmark datasets.
>
> > [1] Sullivan, Gail M., and Richard Feinn. “Using effect size—or why the P value is not enough.” Journal of Graduate Medical Education 4.3 (2012): 279-282.
>
> ---
>
> **3. Comparison with recent baselines:**
>
> We thank your suggestion for a more systematic comparison. To address this, we would like to emphasize that our full Table 12 in Appendix includes a comprehensive comparison of models aiming to improve upon DeiT, including the most relevant baselines we could fairly compare with our approach.
> To strengthen our comparison, we have now included PRepBN[2] and GTP-ViT[3], which were not initially included in our analysis. To ease a clear comparison with our 12-layer DeiT model, we provide a concise overview of GFSA's performance relative to the most relevant and recent baselines, including PRepBN. We believe this addition offers a more systematic evaluation of image classification.
>
> | Method | Top-1 Acc |
> | --- | --- |
> | DeiT-S (12-layers) | 79.8 |
> | + LayerScale | 80.5 |
> | + LateInsertion | 80.5 |
> | + ClassAttention | 80.6 |
> | + AttnScale | 80.7 |
> | + FeatScale | 80.9 |
> | + HAT | 80.9 |
> | + Diverse | 80.6 |
> | + ContraNorm | 80.4 |
> | + NeuTRENO | 80.7 |
> | + *PRepBN* [2] | *80.2* |
> | + *GTP* [3]  | *79.5* |
> | **+ GFSA** | **81.1** |
>
> > [2] Guo, Jialong, et al. "SLAB: Efficient Transformers with Simplified
> Linear Attention and Progressive Re-parameterized Batch Normalization." ICML 2024.
> >
> > [3] Xu, Xuwei, et al. "GTP-ViT: Efficient Vision Transformers via Graph-based Token Propagation." WACV **2024.
>
> ---
>
> ***continued in the next comment***

---

> ### Author Response · Authors · 2024-08-11
>
> **4. Limitations (scalability and extendability):**
>
> We apologize for not fully addressing your concerns about limitations earlier and appreciate the opportunity to clarify these important points.
>
> We conducted additional experiments using the Long Range Arena benchmark[4], which is specifically designed for evaluating performance on long-range dependencies. We tested GFSA on ListOps (2K sequence length) and Image (1K sequence length) datasets using standard Transformers and Efficient Attention [9] models as backbones.
>
> Regarding **scalability** to long sequences, GFSA shows improved performance on these long-sequence tasks. For the Image dataset, Transformer+GFSA outperforms Transformer and some linear Transformers such as Linformer[5], Longformer[6], and YOSO-E[7].
>
> Concerning the **extendability** of efficient/linear Transformers, we successfully integrated GFSA with Efficient Attention[9], demonstrating its compatibility. Efficient Attention+GFSA shows performance gains over Efficient Attention. Importantly, this integration maintains the efficiency benefits of linear transformers. For instance, on the Image dataset, Efficient Attention+GFSA uses only 1.33GB memory and 135.8s/1K-steps, compared to 11.20GB and 737.2s/1K-steps for Transformer+GFSA.
>
> Furthermore, we would like to highlight the potential for efficiency improvements in GFSA. Recent linear complexity Transformers[8,9] propose an approximated self-attention by replacing the softmax operation and rearranging matrix multiplication as $\mathbf{Q(K^\intercal V)}$. In GFSA, the main computational cost comes from calculating $\mathbf{\bar{A}}^2$. However, if we design $\mathbf{\bar{A}}$ as $\mathbf{\bar{A}}=\mathbf{QK}$, then $\mathbf{\bar{A}}^2\mathbf{V} = (\mathbf{QK})(\mathbf{QK})\mathbf{V}$, which allows us to maintain linear complexity even for the squared operation in GFSA by computing it in the same manner as these efficient attention mechanisms. Therefore, the Taylor approximation of the high-order filter in GFSA can be done efficiently.
>
> These results show that our GFSA is scalable to longer sequences and can be effectively extended to efficient/linear transformers. The improvements for longer sequence lengths, along with the efficiency maintained when combined with linear transformers, address concerns about the practical impact of the task.
>
> - Table: Accuracy (%)
>
> |  | ListOps (2K) | Image (1K) |
> | --- | --- | --- |
> | Transformer | 37.1 | 38.2 |
> | **Transformer + GFSA** | **37.6** | **40.2** |
> | Linformer | 37.3 | 37.8 |
> | Longformer | 37.2 | 39.1 |
> | YOSO-E | 37.3 | 39.8 |
> | Efficient Attention | 36.9 | 40.2 |
> | **Efficient Attention + GFSA** | **37.9** | **40.4** |
>
> - Table: Running time (s/1K-steps) and the peak training memory usage (GB)
>
> |  | ListOps (2K) | Image (1K) |
> | --- | --- | --- |
> | Transformer | 198.3/5.50 | 345.1/5.88 |
> | **Transformer + GFSA** | 635.8/ 10.87 | 737.2/11.20 |
> | Linformer | 63.4/1.73 | 158.5/3.45 |
> | Efficient Attention | 49.2/0.57 | 121.1/1.14 |
> |  **Efficient Attention + GFSA** | 53.8/0.67 | 135.8/1.33 |
>
> > [4] Tay, Yi, et al. "Long range arena: A benchmark for efficient transformers." *arXiv preprint arXiv:2011.04006* (2020).
> >
> > [5] Wang, Sinong, et al. "Linformer: Self-attention with linear complexity." *arXiv preprint arXiv:2006.04768* (2020).
> >
> > [6] Beltagy, Iz, Matthew E. Peters, and Arman Cohan. "Longformer: The long-document transformer." *arXiv preprint arXiv:2004.05150* (2020).
> >
> > [7] Zeng, Zhanpeng, et al. "You only sample (almost) once: Linear cost self-attention via bernoulli sampling." ICML 2021.
> >
> > [8] Katharopoulos, Angelos, et al. "Transformers are rnns: Fast autoregressive transformers with linear attention." ICML 2020.
> >
> > [9] Shen, Zhuoran, et al. "Efficient attention: Attention with linear complexities." WACV 2021.
>
> ---
>
> We hope that our additional responses will address your unresolved concerns.
>
> Best regards,
>
> GFSA authors

---

> > ### Author Response · Authors · 2024-08-13
> >
> > Dear Reviewer k2id,
> >
> > As the Reviewer-Author discussion period ends in less than 24 hours, we wanted to reach out to you to see if you are happy with our responses.
> >
> > We have made efforts to address the concerns you raised, particularly:
> > 1. Providing analysis on oversmoothing correlation with DeiT accuracy
> > 2. Clarifying the statistical significance of our improvements
> > 3. Expanding our comparison with recent baselines
> > 4. Addressing limitations in scalability and extendability
> >
> > We believe that your comments have allowed us to improve our paper. We would appreciate your thoughts on whether our extended responses and additional experiments have satisfactorily addressed your concerns. If there are any remaining issues or points that require further clarification, please let us know.
> >
> > Thank you again for your time and dedication in reviewing our work.
> >
> > Kind regards,
> >
> > GFSA Authors

---

> > > ### Author Response · Authors · 2024-08-14
> > >
> > > Dear Reviewer k2id,
> > >
> > > We would like to thank the reviewer once again for taking the time to provide us with valuable feedback, which has enabled us to strengthen our paper with new experiments and clarifications during this rebuttal period.
> > >
> > > As the rebuttal period is nearing its end, we would like to inquire as to whether our additional responses adequately addressed the concerns you raised.
> > >
> > > Furthermore, we would like to express our gratitude for your time and efforts during this rebuttal period. We hope that our responses will allow the reviewer to consider a fresh evaluation of our work.
> > >
> > > Best regards,
> > >
> > > GFSA Authors

---

### Official Review · Reviewer_XKs2 · 2024-07-11

**Soundness:** 3
**Presentation:** 3
**Contribution:** 2
**Rating:** 5
**Confidence:** 4

**Summary:**

This work proposes to enhance self-attention by considering the high-order power of the attention matrix inspired by graph convolution networks and graph signal processing, to overcome the oversmoothing issues of transformers.
To reduce the computational overhead, the authors propose to use a first-order Taylor approximation to the higher-order power of the attention matrix.
With the slightly larger complexity, the proposed technique can improve the performance of Transformers in various fields, across computer vision, natural language processing, etc.

**Strengths:**

1. This paper is well written.
2. The comprehensive empirical experiments well support the advantage of the proposed method and the statement of the work.

**Weaknesses:**

1. The notation in the equation on Line 99 is confusing (the "exponential" in the self-attention matrix).
2. The Taylor approximation well actually hurt the capacity of the graph filters significantly, degenerating to the power of 2, regardless of the value of $K$.
The eq (10) to degenerate to $\hat{\omega}_0 \boldsymbol{I} + \hat{\omega}_1 \bar{\boldsymbol{A}} + \hat{\omega}_2 \bar{\boldsymbol{A}}^2$ for all $K$, where $\hat{\omega}_0, \hat{\omega}_1, \hat{\omega}_2$ are learnable coefficients. Therefore, this technique is just for the power of $2$ and the Taylor approximation actually does not contribute.

**Questions:**

1. Besides the runtime, can you also provide the GPU memory consumption comparisons between GFSAs and the base models?

**Limitations:**

No specific aware.

---

> ### Author Rebuttal · Authors · 2024-08-07
>
> We sincerely thank Reviewer XKs2 for the review and positive feedback, highlighting our strengths in
>
> 1. Well-written paper.
> 2. Comprehensive experiments strongly support the advantage of GFSA.
>
> Below, we would like to address each of your questions and concerns:
>
> **Response to W1: confusing notation**
>
> Thank you for pointing out the notation in the equation on Line 99. We have updated the paper to clarify the exponential notation in the self-attention matrix, which should resolve any confusion.
>
> **Response to W2: Taylor approximation**
>
> We appreciate your careful consideration of our method. We respectfully disagree with the view that our approach degenerates to only second-order terms and would like to provide a more detailed explanation of our Taylor approximation approach.
>
> The Taylor approximation in our work is specifically used to estimate higher-order terms ($\bar{\boldsymbol{A}}^K$ for $K > 2$) efficiently. It is not intended to limit the model to second-order interactions but rather to capture the essence of higher-order terms without incurring computational costs. As detailed in Eqs. (6)-(8) of our paper, we use a first-order Taylor approximation at point $a = 1$, combined with a forward finite difference method to approximate the derivative. This results in the approximation: $\bar{\boldsymbol{A}}^K ≈ \bar{\boldsymbol{A}} + (K-1)(\bar{\boldsymbol{A}}^2 - \bar{\boldsymbol{A}})$. This formulation includes both $\bar{\boldsymbol{A}}$ and $\bar{\boldsymbol{A}}^2$ terms, but importantly, it scales with $K$, allowing it to capture aspects of higher-order interactions.
>
> In Eq. (10), our GFSA filter is defined as $H_{GFSA} = w_0 \boldsymbol{I} + w_1 \bar{\boldsymbol{A}} + w_K(\bar{\boldsymbol{A}} + (K-1)(\bar{\boldsymbol{A}}^2 - \bar{\boldsymbol{A}}))$. The learnable coefficients $w_0$, $w_1$, and $w_K$ provide the model with the flexibility to balance the contributions of different order terms. Notably, $w_K$ scales the approximated higher-order term, allowing the model to adjust its influence based on the task requirements. Tables 8, 10, and 11 show improved performance as $K$ increases. This improvement would not be observed if the model were limited to second-order interactions.
>
> This approximation allows us to capture higher-order effects without the full computational burden of calculating $\bar{\boldsymbol{A}}^K$ directly for large $K$, which would be expensive for many practical applications. While our approach does use a Taylor approximation to estimate higher-order terms, it is not equivalent to a simple second-order polynomial. The approximation retains a dependency on $K$, allowing it to capture aspects of higher-order interactions in a computationally efficient manner.
>
> **Response to Q1: GPU Memory**
> Thank you for your question regarding GPU memory consumption. We have measured GPU memory usage (GB) during training for image classification and natural language understanding tasks. The results are presented in the table below.
>
> The increase in GPU memory consumption can be attributed to the calculation of $\bar{\boldsymbol{A}}^2$ in our GFSA method. However, it's crucial to consider this increase in the context of GFSA's overall impact on model performance and efficiency. GFSA adds only about 100 parameters to the model, yet yields improvements across various tasks.
>
> Moreover, our 12-layer DeiT+GFSA outperforms 24-layer DeiT, which has significant implications for computational efficiency. As detailed in Table 20, 24-layer DeiT requires approximately 50 minutes per epoch, while 12-layer DeiT+GFSA needs only about 30 minutes per epoch yet achieves higher accuracy. This demonstrates that GFSA can achieve better results with lower overall memory requirements and reduced runtime compared to simply increasing the number of layers in the base model.
>
> Considering these factors - the performance improvements, the minimal increase in parameters, and the potential for reduced runtime - we believe the additional GPU memory usage is acceptable.
>
> | Model  | 12-Layer | 24 Layer |
> | --- | --- | --- |
> | DeiT-S | 1.6 | 3.2 |
> | DeiT-S+GFSA | 2.1 | 4.0 |
>
> | Model | CoLA | SST2 |  MRPC |  QQP |  STSB |  MNLI | QNLI | RTE | Avg |
> | --- | --- | --- | --- | --- | --- | --- | --- | --- | --- |
> | BERT$_{BASE}$ | 2.31 |  2.73 |  3.73 |  4.42 |  4.33 |  4.45 |  4.45 | 4.45 | 3.86 |
> | BERT$_{BASE}$   + GFSA |  2.39 |  2.90 |  4.13 | 5.00  |  4.86 |  4.98 | 5.00 |  4.98 | 4.28 |
> | ALBERT$_{BASE}$ |  1.69 |  2.89 |  4.43 |  4.56 |  4.55 |  4.57 |  4.57 |  4.57 |  3.98 |
> | ALBERT$_{BASE}$ + GFSA|  1.77 |  3.15 |  5.01 |  5.14 |  5.12 |  5.14 |  5.14 |  5.14 |  4.45 |
> | RoBERTa$_{BASE}$ |  2.48 |  2.96 |  3.91 |  4.60 |  4.49 |  4.59 |  4.60 |  4.62 |  4.03 |
> | RoBERTa$_{BASE}$ + GFSA | 2.55 | 3.13 | 4.30 | 5.19 | 5.05 |  5.17 |  5.20 |  5.15 |  4.47 |

---

> ### Author Response · Authors · 2024-08-08
>
> Dear reviewer XKs2,
>
> We would like to provide additional clarification to address your concern about our Taylor approximation method.
>
> To further demonstrate that our GFSA does not degenerate to second-order terms and that the Taylor approximation contributes meaningfully, we would like to emphasize the empirical result we have already provided in our paper, specifically in Appendix S, Table 37.
>
> For your convenience, as shown in Table 37 of the paper here:
> | Datasets | #Params | CoLA | SST-2 | MRPC | QQP | STS-B | MNLI-m/mm | QNLI | RTE | Avg |
> |----------|---------|------|-------|------|-----|-------|-----------|------|-----|-----|
> | BERT_BASE | 110M | 56.79 | 93.81 | 88.70 | 88.32 | 88.16 | 84.96/84.15 | 91.63 | 66.06 | 82.51 |
> | + GFSA (approximated $\mathbf{\bar{A}}^K$) | 110M | 59.56 | 94.15 | 90.60 | 88.46 | 88.33 | 85.12/85.06 | 91.95 | 68.95 | 83.58 |
> | + GFSA (actual $\mathbf{\bar{A}}^K$) | 110M | 59.85 | 94.27 | 89.80 | 88.43 | 88.32 | 84.95/84.89 | 91.76 | 68.23 | 83.39 |
>
> This data clearly shows that our GFSA method with the approximated $\mathbf{\bar{A}}^K$ achieves performance very close to that of using the actual $\mathbf{\bar{A}}^K$. The average performance across all GLUE tasks is higher for the approximated $\mathbf{\bar{A}}^K$ compared to the actual $\mathbf{\bar{A}}^K$.
>
> These results successfully relieve the reviewer's concern that our approach degenerates to only second-order terms. If this were the case, we would expect to see a significant performance drop when using the approximated $\mathbf{\bar{A}}^K$ compared to the actual $\mathbf{\bar{A}}^K$. Instead, we observe comparable or even slightly improved performance. Additionally, the approximation allows us to balance computational efficiency with model expressiveness.
>
> We believe this empirical evidence, along with our previous response, addresses your concern about the effectiveness of our Taylor approximation approach. Please let us know if you have any remaining concerns, and we will happily respond. Thank you!
>
> Best regards,
>
> GFSA authors

---

> > ### Comment · Reviewer_XKs2 · 2024-08-12
> >
> > Thank you for the rebuttal.
> > The rebuttal addresses most of my concerns.
> >
> > One thing I am not certain about is the gain and complexity trade-off. The proposed method consistently shows improvements on various tasks, with a larger computational complexity $O(N^3)$.
> >
> > In practice, based on runtime and memory provided, the extra computational cost is acceptable. However, the performance improvement is not remarkable either.
> >
> > Considering my current score, I tend to retain the current score for now and would like to decide whether to raise the score during the reviewer and AC discussion stage.

---

> > > ### Author Response · Authors · 2024-08-13
> > >
> > > Dear reviewer XKs2,
> > >
> > > We thank you for considering the gain and complexity of the trade-off. We want to address your concerns about our complexity from 3 perspectives below:
> > >
> > > **[GFSA in selective layers]**
> > >
> > > Regarding the computational overhead of GFSA, we introduced a strategy for computational complexity in Section 6. By applying GFSA only to even layers, we effectively reduced the runtime increase while maintaining performance similar to that of full-layer integration.
> > >
> > > **[Possibility of improving complexity by using an efficient matrix operation]**
> > >
> > > As we mentioned in Appendix P, our GFSA computes $\bar{\mathbf{A}}^2$, which has a time complexity of $O(n^2d + n^3)$.  However, assuming we use the algorithm in [1], the time complexity becomes $O(n^2d + n^{2.371552})$. In practical terms, if $n^{0.371552}$ is smaller than $d$, the time complexity of GFSA approaches $O(n^2d)$.
> > >
> > > In real-world applications, especially those leveraging GPU acceleration, the practical computational cost of matrix operation could be better than the theoretical bound. Because GPUs with multiple CUDA cores provide acceleration for matrix operations. This means that the actual wall-clock time for these computations can be lower than the theoretical time complexity.
> > >
> > > **[Additional results for GFSA with linear complexity]**
> > >
> > > We want to share **additional results** that may address your concern about the computational complexity, which we reported in our response to Reviewers `k2id` and `bzWT`. While our method has $O(n^2d + n^3)$ complexity due to the nature of self-attention, we have found that GFSA can be applied with linear complexity calculations while retaining its benefits.
> > >
> > > In experiments using Long Range Arena benchmark [2], we successfully integrated GFSA with Efficient Attention [3], an approach that maintains linear complexity. The results show that Efficient Attention+GFSA outperforms Efficient Attention alone while preserving efficiency. For example, on Image dataset (1K sequence length), Efficient Attention+GFSA uses only 1.33GB memory and 135.8s/1K-steps, compared to 11.20GB and 737.2s/1K-steps for Transformer+GFSA.
> > >
> > > | Method | ListOps (2K) | Image (1K) |
> > > | --- | --- | --- |
> > > | Transformer | 37.1 | 38.2 |
> > > | **Transformer + GFSA** | **37.6** | **40.2** |
> > > | Linformer | 37.3 | 37.8 |
> > > | Longformer | 37.2 | 39.1 |
> > > | YOSO-E | 37.3 | 39.8 |
> > > | Efficient Attention | 36.9 | 40.2 |
> > > | **Efficient Attention + GFSA** | **37.9** | **40.4** |
> > >
> > > > Note: Results are shown as Accuracy (%)
> > >
> > > |  | ListOps (2K) | Image (1K) |
> > > | --- | --- | --- |
> > > | Transformer | 198.3/5.50 | 345.1/5.88 |
> > > | **Transformer + GFSA** | 635.8/ 10.87 | 737.2/11.20 |
> > > | Linformer | 63.4/1.73 | 158.5/3.45 |
> > > | Efficient Attention | 49.2/0.57 | 121.1/1.14 |
> > > |  **Efficient Attention + GFSA** | 53.8/0.67 | 135.8/1.33 |
> > >
> > > > Note: Results are shown as Running time (s/1K-steps) / Peak training memory usage (GB)
> > >
> > > We also found a way to maintain linear complexity even for the squared operation in GFSA. By designing $\mathbf{\bar{A}}$ as $\mathbf{\bar{A}}=\mathbf{QK}$, we can efficiently compute  $\mathbf{\bar{A}}^2\mathbf{V} = (\mathbf{QK})(\mathbf{QK})\mathbf{V}$, similar to recent linear complexity Transformers [3,4].
> > >
> > > These findings show that GFSA can be implemented with linear complexity and still provide performance improvements. This addresses the concern about the computational cost and shows that the gain-complexity trade-off can be more favorable than initially presented.
> > >
> > > > [1] Williams et al. New bounds for matrix multiplication: from alpha to omega. In SODA, 2024.
> > > >
> > > > [2] Tay, Yi, et al. "Long range arena: A benchmark for efficient transformers." *arXiv preprint arXiv:2011.04006* (2020).
> > > >
> > > > [3] Shen, Zhuoran, et al. "Efficient attention: Attention with linear complexities." WACV 2021.
> > > >
> > > > [4] Katharopoulos, Angelos, et al. "Transformers are rnns: Fast autoregressive transformers with linear attention." ICML 2020.
> > >
> > >
> > > ---
> > >
> > > We hope that our responses address your concerns and that you will be able to increase your rating. If you have any questions, please let us know.
> > >
> > > Best regards,
> > >
> > > GFSA authors

---

### Official Review · Reviewer_Py4c · 2024-07-12

**Soundness:** 3
**Presentation:** 3
**Contribution:** 2
**Rating:** 4
**Confidence:** 4

**Summary:**

The paper proposed a graph-filter-based self-attention mechanism to improve its effectiveness. The authors conduct experiments in various fields, including natural language understanding, computer vision, automatic speech recognition, graph regression, and code classification, showing the generalization of the proposed method. The proposed method is claimed to make a difference in the over-smoothing problem, where representations across layers converge to indistinguishable values, leading to significant performance degradation.

**Strengths:**

1. Sufficient Experiments. The paper conducts experiments on natural language understanding, computer vision, automatic speech recognition, graph regression, and code classification. The authors try their best to show the generalization of the proposed method.
2. Well Organized. The paper articulated the viewpoint in a clear and organized manner.
3. Reproducibility. The paper shows the pseudo-code, which clearly presents the implementation of GFSA in the appendix. The paper shows detailed experiment settings in appendices I to O. The code is available with instructions. However, an error occurred when I tried to run it, which is reported in the following questions.

**Weaknesses:**

1. Sensitivity of K. The authors did not give instructions on selecting the value of K. The sensitivity of K is discussed in appendices, showing that 'For each dataset, there is optimal K and the performance of models using GFSA is generally robust to changes in K'. All the experiments searched for values of K. However, the search for K is time-consuming.
2. Misleading Figure. Fig 1 shows the performance of the proposed method. However, the inconsistent proportions exaggerate the model's effectiveness.

**Questions:**

1. Miss code. When I was running the experiment of code classification, using the command “python run_exp.py --model_tag codet5_base --task defect --sub_task none” as instructed in the readme file, an error occurred reporting ‘can't open file '/run_gen.py'’ . It seems that some code is missing.
2. More questions about the sensitivity of K. Is it a necessary step to search the optimal K when facing a new dataset? Or is it robust in K? I noticed that, in Theorem 4.1, Ek is relevant to K.

**Limitations:**

The authors addressed the limitations, including effectiveness and efficiency.

---

> ### Author Rebuttal · Authors · 2024-08-07
>
> We sincerely thank Reviewer Py4c for the review and  feedback, highlighting our strengths in
> 1. Extensive experimentation across diverse domains demonstrating the generalizability of GFSA.
> 2. Clear and well-organized presentation of ideas.
> 3. Strong emphasis on reproducibility with detailed implementation information and code availability.
>
> Below, we would like to address each of your questions and concerns:
>
> **Response to W1: Sensitivity of $K$**
>
> We understand your concern about the time required to find the optimal $K$. Our tasks across various domains include both fine-tuning and training from scratch. For fine-tuning tasks with relatively short training times, such as the GLUE benchmark, searching for $K$ values from 2 to 10 is not a significant hindrance (see Appendix I.2). However, for tasks like ImageNet that require training from scratch, we flexibly set a narrower search space of 2 to 5 for the 12-layer DeiT-S model.
>
> **Response to W2: Misleading figure**
>
> We acknowledge that the different scales and metrics for each task could potentially lead to misinterpretation. We have redrawn the plot based on specific backbones, showing only the percentage of improvement. We hope that the updated Figure 1 in the pdf file of the general response addresses this concern and eliminates any potential for misleading information.
>
> **Response to Q1: Missing code**
>
> We apologize for the inconvenience. We have uploaded the `run_gen.py` file, and the experiment should now run successfully. Please check the same link provided in the paper for the updated code repository.
>
> **Response to Q2. Sensitivity of $K$**
>
> Regarding the necessity of searching for the optimal $K$ for new datasets, our experiments suggest that while there is an optimal $K$ for each dataset, the performance of models using GFSA is generally robust to changes in $K$. However, for best results, we recommend conducting a search for the optimal $K$ when dealing with a new dataset.
> As for the relation between $K$ and $E_k$ in Theorem 4.1, you are correct in noticing this connection. The theorem provides a theoretical foundation for understanding how $K$ affects the model's performance. While the model shows robustness to $K$ in practice, the optimal value can vary depending on the specific dataset and task.
>
> Importantly, we would like to draw your attention to Table 8 in Appendix J.2, which demonstrates the effectiveness of GFSA across different $K$ values. This table shows that for the three datasets, all tested $K$ values (from 2 to 9) result in better performance compared to the original GPT2 model. This provides evidence that GFSA consistently improves performance across a range of $K$ values. This robustness to $K$ is a strength of our method.

---

> > ### Author Response · Authors · 2024-08-13
> >
> > Dear Reviewer Py4c,
> >
> > As the Reviewer-Author discussion period ends in less than 24 hours, we wanted to reach out regarding our response to your valuable feedback. Please let us know if you would like us to help with any further details on our response.
> >
> > We believe that your comments have allowed us to improve our analysis of the sensitivity of $K$ and address potential misunderstandings in Figure 1. In addition, we have addressed concerns and questions about the sensitivity of K.
> >
> > Additionally, we would like to draw your attention to our response to other reviewers, `k2id` and `XKs2`, where we discussed **the scalability of GFSA and its extendability to linear/efficient Transformers**. We believe these findings might be of interest to you as they further show the broader applicability and potential of our method.
> >
> > Thank you again for your time and dedication in reviewing our work!
> >
> > Kind regards,
> >
> > GFSA Authors

---

> > ### Author Response · Authors · 2024-08-14
> > **A gentle reminder**
> >
> > Dear Reviewer Py4c,
> >
> > We thank the reviewer again for your time and feedback that allowed us to strengthen the paper with clarifications during this important rebuttal period.
> >
> > As the end of the rebuttal period is fast approaching we were wondering if our answers in the rebuttal were sufficient enough to address the important concerns.
> >
> > Finally, we are very appreciative of your time and effort in this rebuttal period and hope our answers are enough for the reviewer to consider a fresher evaluation of our work with a potential score upgrade if it's merited.
> >
> > Kind regards,
> >
> > GFSA Authors

---

> ### Comment · Area_Chair_7dbq · 2024-08-12
>
> Hello, Reviewer. The author has submitted a response to your comments. Whether or not it addresses your concerns, it would be greatly appreciated if you could acknowledge that you have reviewed the reply.

---

### Author Rebuttal · Authors · 2024-08-07

Dear reviewers,

We sincerely appreciate your feedback and constructive comments on our paper. We are grateful for the recognition of several key strengths in our work:
1. Extensive experiments across diverse domains demonstrating the broad applicability of GFSA
2. Clear, well-organized presentation of ideas
3. Emphasis on reproducibility with detailed code and settings
4. Versatility of GFSA across different transformer variants and fields
5. Innovative application of graph filter concepts to transformers
6. Easily graspable core idea facilitating adoption
7. Comprehensive empirical support for the advantages of GFSA

**Important: We have uploaded a one-page PDF file with additional materials addressing your concerns.**


This pdf includes:
- Updated Fig. 1: The original Fig.1 was potentially misleading because different metrics for different tasks were shown in the same figure, so we changed the radar chart to show the improvement ratio compared to the backbone. This change is related to W2 of reviewer Py4c.
- New Tables 1 and 2: To address the weakness (W2) of reviewer k2id that only two datasets were considered for the tasks in the graph domain, we include additional experimental results applying GFSA to GraphGPS and Graph-ViT. We provide more rigorous evaluations of GFSA by adding 7 datasets: Peptide-Func, Peptide-Struct, ZINC, MNIST, CIFAR10, Molhiv, and MolTOX21.
- New Fig. 2:  The analysis of filter responses based on the learned coefficients of GFSA across the different layers of BERT is shown. As the layer deepens, there is a noticeable shift towards higher frequency responses, indicating a move towards high-pass filtering. Therefore, Fig. 2 provides the evidence for GFSA's effectiveness in preserving high-frequency information. This figure is related to W4 of reviewer k2id.

We encourage you to review this material as it provides visualizations and additional results that may enhance your understanding of our responses to your questions and concerns.

---

### Author Response · Authors · 2024-08-14

Dear Reviewers and ACs,

We sincerely appreciate the constructive feedback from all reviewers. Based on your valuable comments, we have made the following improvements:

1. Extended our experiments with recent models:
    - Added results for GPS and Graph-ViT on additional datasets from LRGB and Benchmarking GNNs.
    - Demonstrated consistent improvements across various datasets, further validating the effectiveness of GFSA.
2. Clarified and refined our theoretical analysis:
    - Provided a more rigorous proof for Theorem 3.1.
    - Offered a detailed explanation of how learned coefficients determine filter characteristics.
3. Conducted additional analyses:
    - Visualized frequency responses across layers to demonstrate GFSA's adaptive filtering behavior.
    - Provided GPU memory usage comparisons between backbone models and GFSA.
4. Addressed scalability and efficiency concerns:
    - Demonstrated GFSA's compatibility with efficient/linear Transformers such as Efficient Attention.
    - Showed performance improvements while maintaining efficiency on long-sequence tasks.

After our rebuttal, 2 reviewers expressed satisfaction with our responses:

- Reviewer bzWT **flipped the decision from reject to borderline accept**, recognizing the positive evidence for learning GFSA filters.
- Reviewer XKs2 acknowledged that we addressed most of their concerns, **maintaining the positive evaluation** of our submission.

We sincerely appreciate the further engagement from these reviewers.

However, reviewer Py4c has not yet responded to our rebuttal, and reviewer k2id has not yet responded to our additional rebuttal. We believe that we have satisfactorily addressed all concerns from reviewers Py4c and k2id.

Again, we would like to thank you and all the reviewers for your great efforts in reviewing our paper. If reviewers are satisfied with our response or have any more concerns, please do not hesitate to let us know.

Best regards,

GFSA Authors

---

### Decision · Program_Chairs · 2024-09-25

**Decision:**

Accept (poster)

**Comment:**

This work proposes enhancing self-attention by considering the higher-order powers of the attention matrix, inspired by graph convolutional networks and graph signal processing, to address the oversmoothing issues of transformers. To reduce computational overhead, the authors suggest using a first-order Taylor approximation of the higher-order powers of the attention matrix. With a slightly larger computational complexity, the proposed technique can improve the performance of transformers in various fields.

Reviewer Py4c points out that a hyperparameter, K (power), needs to be determined, and that searching for K can be computationally expensive. The authors provide evidence that GFSA consistently improves performance across a range of K values. Since K is robust, this concern is addressed.

Reviewer XKs2 pointed out that the Taylor approximation may significantly reduce the capacity of the graph filters. The authors address this issue from both the theoretical and experimental perspectives. Reviewer XKs2 also has concerns about the trade-off between gain and complexity. The proposed method consistently shows improvements on various tasks but with a higher computational complexity. The authors offer some ideas to reduce the computational requirements of the proposed approach.

Reviewer k2id is concerned about the weak baseline and provides several recent methods for comparison. The rebuttal addresses this concern. The authors point out that the recent methods suggested by the reviewer are already included in the paper, but they are in the appendix.

Reviewer bzWT has concerns about novelty, which are addressed during the rebuttal. Reviewer bzWT also has some questions and identifies some issues regarding Theorem 3.1. The authors promise to provide a more rigorous proof for Theorem 3.1.